# Sociotechnical imaginaries of autonomous vehicles: Comparing laboratory and online eye-tracking methods

Mergime Ibrahimi [1,2]*, Anu Masso[1,2], Mauro Bellone[1]

1 FinEst Centre for Smart Cities, Tallinn University of Technology, Tallinn, Estonia, 2 Ragnar Nurkse Department of Innovation and Governance, Tallinn University of Technology, Tallinn Estonia

☯ These authors contributed equally to this work.
* mergime.ibrahimi@taltech.ee

## Abstract

This study investigates sociotechnical imaginaries of autonomous vehicles (AVs) using a dual approach: in-lab and online eye-tracking experiments. We examine how cognitive engagement varies across hypothetical decision-making scenarios involving algorithmic failure of AVs. In comparison with non-AV scenarios. This article highlights the characteristics, advantages, and limitations of methods, emphasizing their complementary contributions to understanding how individuals perceive and engage with emerging technologies. The in-lab experiment revealed high-quality and precise data from a homogeneous sample, while the online experiment enabled us to scale the research and explore diverse sociotechnical imaginaries from a global sample through crowd-sourced platforms. Key findings show that both in-lab and online participants exhibited longer gaze durations at one point, predominantly longer in AV scenarios. However, a deeper analysis of overall cognitive engagement revealed that in-lab participants, with more concentrated sociotechnical imaginaries, were more focused on non-AV scenarios, indicating a stronger emphasis on human decision-making. In contrast, online participants, whose imaginaries may be shaped by global perspectives and diverse experiences with data and algorithms, displayed increased attention toward AV scenarios, with significant visual variations among participants, reflecting global interest or concern over high-stakes algorithmic decisions. These findings contribute to our understanding of how perception of AVs differs globally and offer insights into emerging concerns around algorithmic decision-making in everyday life.

## 1. Introduction

Autonomous vehicles steer us into the future and supposedly solve major mobility problems – to assure access to the necessary infrastructure in the cities, and

**Data availability statement:** Data cannot be shared publicly because they are sensitive and confidential. However, to ensure the confidentiality and anonymity of the respondents—as recommended by the research ethics committee—we have taken specific measures to balance the openness of science with the protection of research subjects. This is particularly important given the context of our study: a small-scale study with a limited number of participants who could potentially be identified, and a large-scale study addressing the evolving international regulations concerning platform workers' rights. To achieve this balance, we have taken the following concrete steps: 1. Meta-data descriptions: We have prepared thorough metadata descriptions to provide detailed explanations of the methodology used in the study. These metadata are openly available here: https://doi.org/10.5281/zenodo.13919434. 2. Data access request process: Researchers who wish to reuse the data can request access from Henri Schasmin (henri.schasmin@taltech.ee), who is the Coordinator for Protection of Personal Data at Tallinn University of Technology and is responsible for ensuring that all handling of the dataset complies with institutional and legal data protection standards. In cases of data requests, a confidentiality agreement will be established between the researchers, in line with the data processing principles required by the research ethics committee. We are committed to finding solutions that uphold the ideals of open science while protecting the rights and privacy of research participants.

**Funding:** European Union's Horizon 2020 Research and Innovation Programme under grant agreement No. 856602 (Finest Twins), Development program ASTRA of Tallinn University of Technology for years 2016-2022 (2014-2020.4.01.16-0032), and European Union's Horizon Europe Research and Innovation Programme under grant agreement No.~101135988 (PLIADES: AI-Enabled Data Lifecycles Optimization and Data Spaces Integration for Increased Efficiency and Interoperability).

**Competing interests:** The authors have declared that no competing interests exist.

therefore, create imaginaries of a perfect future. This article aims to investigate sociotechnical imaginaries on these futures, through providing a deeper understanding of our collective stance toward autonomous vehicles, by comparing the in-lab and online eye-tracking methods.

The use of data and artificial intelligence in planning smart cities, including collecting data from public spaces, can have a big effect on how cities are developed, managed, viewed, and utilized [1]. One of the most controversial fields where data are used for planning better mobility solutions is the introduction of autonomous vehicles (AVs). However, how people understand autonomous vehicles is often not reflected, even though the peculiarities of people are embedded in data that are used to train the algorithm of AVs. Thus, it is important to understand the sociotechnical imaginaries – the envisaging of desirable futures attained through science and technology [2] – of the future of transport systems and mobility by analyzing it from the human perspective. In this study, we use AVs as a proxy to explore sociotechnical imaginaries through eye-tracking, however, the methods used in this study can be applied to examine similar dynamics in other domains.

We consider autonomous vehicles as an automated data technology system since they run without the mediation of a person [3], but instead are mediated through algorithms, and take the role of an automated decision-maker. Throughout this paper, the term AVs will refer to SAE level 5 automated system – the vehicle can take control of the vehicle under all conditions. In this study, we adopt the perspective that algorithms are not natural entities but socially constructed – designed by humans and trained based on human-generated data. Given this understanding of algorithms, it is crucial to understand how social perceptions are shaped by algorithmic technologies.

There is a growing body of academic literature that studies public perceptions and imaginaries toward automated decision-making systems. A study by Liu, Yang and Xu [4] looked at public acceptance of self-driving cars, examining factors that affect attitudes such as trust, perceived risks and benefits, and environmental concerns. A study conducted by Pudāne, Van Cranenburgh and Chorus [5] demonstrated that the adoption of fully AVs significantly reshapes daily activities, particularly by increasing multitasking during travel, with more pronounced effects among higher-income and well-educated users. Similarly, Li et al. [6] used structural equation modeling to examine the factors influencing human drivers' trust in highly autonomous vehicles, revealing that perceived safety and system reliability are key determinants of trust, which in turn affects the willingness to adopt this technology. Additionally, Chen et al. [7] explored the association between socio-demographic factors and public acceptance of fully autonomous vehicles, highlighting that factors such as age, income, and education significantly influence attitudes toward these technologies.

Nevertheless, there is a notable gap in the literature on scrutinizing how diversity is transformed in data, particularly how social diversity is transformed into automated technologies. Data tends to be seen "without critically reflecting on its own mediation, implicit values and its performativity" [[8], p.2]. We assume that disregarding diversity in automated technologies may lead to many societal problems in mobility, such as biases towards certain groups of pedestrians or even traffic fatalities if AVs fail to

detect certain people. Despite the enthusiastic speculation of AVs, there is a methodological gap to empirically study the imaginaries of AVs, particularly regarding social diversity and the implicit understanding of this technology.

This study extends previous research by using a mixed and experimental approach to study the sociotechnical imaginaries of automated technologies. The starting point of this study is the Moral Machine (MM) experiment by Awad et al. [9] aimed to understand how people from different cultures and backgrounds would make ethical decisions in scenarios involving AVs. Building on this foundation, we developed the experiment further by studying the sociotechnical imaginaries of AVs through eye-tracking technology – a technology that enables to analyze the perception of people when they interact with specific stimuli. We hypothesize, based on previous research [10–12], that there is a significant difference in cognitive engagement when people observe autonomous vehicles compared to non-autonomous vehicles in high-stakes situations such as vehicle brake failure – choosing between passengers or pedestrians. We assumed that this difference might stem from participants' experience and knowledge regarding emerging technologies, such as data and algorithms, which could influence their perception and engagement with AVs.

The new aspect of our research lies in introducing eye-tracking as a valuable method for examining the unknown cognitive mechanisms involved in how people perceive emerging technologies. Furthermore, this paper provides a direct comparison between laboratory (in-lab) and online-based eye-tracking experiment methods. While the use of crowd-sourced and online platforms to measure cognitive engagement is on the rise, little is known about how these methods compare to traditional in-lab approaches. Our study addresses this gap by having participants complete identical scenarios in both settings, allowing for a unique evaluation of the strengths and differences between these two experimental approaches.

## 2. Literature review

### 2.1. Sociotechnical imaginaries on autonomous vehicles

Studies agree on the importance of sociotechnical imaginaries – analyzing values and norms as means for better understanding the accessibility to the infrastructure in the smart cities [13]. However, biased images of urban processes lean towards corporate and instrumental visions of the city rather than including and reflecting public values and citizens' interests [14]. Therefore, it is necessary to better understand the peoples' perceptions to reveal the citizens' interests in smart city data-based developments.

This article uses the definition of sociotechnical imaginaries by Jasanoff [2] who takes into consideration the complexities of future vision attained through science and technology, "collectively held, institutionally stabilized, and publicly performed visions of desirable futures, animated by shared understandings of forms of social life and social order attainable through, and supportive of, advances in science and technology" [2, p.6]. Therefore, sociotechnical imaginaries allow us to better understand the interplay of technology and the societal aspects – potential consequences, benefits, and risks through the perceptions of the individuals.

Prior studies on the socio-technical imaginaries have emphasized that algorithms used in automated decisions often tend and are seen to disregard data diversity as social sorting is quite prominent in algorithmic selections. For example, some studies examined biases and inequalities of algorithmic calculations [15] by foremost exploring how vulnerable social groups are neglected by algorithms, like ethnic minorities, older people, and people with disabilities, and the intersections of these [16]. It is argued that technologies that rely on data to make automated decisions can yield biased and discriminatory results [17]. However, the bias in data is not generated by the discrimination that exists in society, as those data can be filtered, but rather from the intrinsic characteristics of statistical algorithms.

There is a wide array of studies about the imaginaries and techno-visions of autonomous vehicles, such as getting information about real responses through simulated worlds [18], or using controversies to help people identify their own issues and imagine alternative possibilities [8]. In literature, it is argued that the imagination of autonomous vehicles is determined by the moral beliefs of one's society or culture based on what is morally right, wrong, good, bad, etc. [19].

Therefore, prior studies on the imaginaries have mostly focused on the ethical and moral dilemmas, like studying the evolution of moral and ethical choices [20], the ethics of accidents with autonomous vehicles, and the moral choices in designing and implementing such algorithmic solutions [19], and individuals being often powerless in datafication and developing data-based smart cities [1].

Other studies have explored the perceptions of algorithmic threats across socio-demographic variables [21,22], including age – younger people being more tolerant, whereas gender and education did not play a significant role, gender biases and stereotypes [20]. A study conducted by Nagtegaal [23] found that people perceive human decisions as fairer in high-complexity tasks, while favoring algorithmic ones in low-complexity tasks. On the other hand, Araujo et al. [10] found a preference for algorithmic decisions in high-stakes health and justice decisions. Another study found that people assigned less responsibility to self-driving cars at fault compared to human drivers, potentially influenced by media promoting AVs as solutions to current mobility challenges [11,12]. It is argued that such sociotechnical imaginaries create a future-perfect narrative, sidestepping critical questions about the social impacts of disruptive technologies [24].

## 2.2. Investigating sociotechnical imaginaries on autonomous vehicles

Several research methods have been used to measure people's perceptions of algorithms, including surveys [10,22], interviews [25], and the Technology Acceptance Model [26], which rely on explicit data —self-reported information that reflects conscious thoughts and attitudes— that may sometimes be insufficient. While these approaches offer advantages, such as capturing subjective views based on life experiences, they can be imprecise because there may be a discrepancy between what people say and what they do (for real-time vs. retrospective, subjective measurements, see [27,28]). Cognitive methods like eye-tracking and EEG offer a significant complement to traditional approaches by providing precise data on visual attention and uncovering hidden, implicit interactions between people and algorithmic systems [29] improvised in a digital form. Our eyes follow what interests us, and our brains are active when we are intrigued or emotionally engaged by what we observe.

Recently, there has been a rising interest in interdisciplinary approaches, like combining anthropological and computational approaches [30], and a need for new methodologies in studying machine behaviour [31], such as the methodological twin-move of making big data thick and thick data big [32]. These studies have explored how autonomous vehicles will "see" to navigate themselves and, more importantly, avoid colliding with people, other vehicles, and objects [18], trust and threats of autonomous machines [33], liability and responsibility of autonomous vehicles [34], and pedestrians' acceptance [22]. Additionally, the relationship between driver trust and attention during autonomous driving has been shown to play a crucial role in traffic safety and public acceptance. For example, a study by Li et al. [35] found that while latent hazard notifications improve driver attention and safety by increasing focus on potential risks, they also boost driver trust, which paradoxically reduces attention and diminishes 15.12% of the expected safety benefits due to behavioral adaptation.

However, there are methodological difficulties and gaps in measuring implicit beliefs —subconscious attitudes and perceptions that individuals may not be aware of— and stereotypes towards algorithms [36]. Therefore, there is an increasing interest in eye-tracking, VR, and other experimental tools and cognitive research methods [37,38], which allows researchers to analyze visual behaviors to better understand the implicit and hidden understandings of people towards data technologies. Eye-tracking enables exploration of human cognitive capabilities [39] and turning the process on the other way around – using the cognitive tools and human perception to understand the machines, algorithms, and AI, and how to engage the citizens in datafied smart city planning [8].

One of the ways for operationalizing and measuring sociotechnical imaginaries and people's cognitive perceptions, particularly in the context of technology, involves the exploration of moral decision-making processes. Alan Turing already in the '50s suggested the "imitation game" proposing a method to check if a machine is intelligent compared to a human, introducing the controversial topic about how/if a machine's intelligence can be measured via a method later called the "Turing test" [40]. However, whether autonomous vehicles include explicit moral decision-making is still unknown since

coding morals into machines is not an easy task [41]. Nevertheless, autonomous vehicles involve social risk, which constitutes an ethical decision [42]. Any current or near-future autonomous system lacks the wealth of knowledge people use to make moral decisions, such as diverse characteristics of victims or subtle features of the environment. However, a car's actions in road traffic can have moral consequences, and if an algorithm faces a situation with moral consequences, it is possible that its actions would be considered motivated by morals [41]. But then again, how could 'morals' be coded?

Additionally, scenarios are widely used to understand and respond to the impacts of human activities and to describe potential alternative futures [43]. Picture scenarios, in particular, help designers and decision-makers explore human behavior and interactions with different outcomes. This method sparks imagination and provides new perspectives in uncertain futures. Studies have shown that visual discourse analysis reveals how pictures are used as a medium to construct and disseminate sociotechnical imaginaries [12]. For example, Kastner [44] used eye-tracking technology to explore the relationship between people's moral judgments about morally challenging situations and their visual attention to related images.

As many experimental tasks, including the scenarios, are performed on computers, scholars have conducted online studies instead of having participants come to a lab [45,46]. Although there are obvious limitations, such as reduced experimental control and data quality when experiments are run online, some human-computer interaction research can reliably be conducted online as it enables the possibility to scale up the study and compare data across different participants from different locations and backgrounds [47–49]. Despite the development of in-lab and online experiments, we still do not know about the differences, benefits, obstacles, and disadvantages of such experimental settings.

## 3. Methods and data

This article explores the characteristics, advantages, and limitations of both in-lab and online experiments, highlighting how each approach contributes to the scientific understanding of using eye-tracking technology in the social sciences and investigating sociotechnical imaginaries towards algorithms in daily transport situations.

To do that, we conducted an in-lab and online experiment by using a mixed-method approach to assess people's imaginaries of autonomous vehicles. More in detail, we 1) constructed original interactive scenarios in the form of visual narratives (inspired by the MM experiment by Awad et al. [9] and developing it further to meet the previously mentioned criticism of the methods); 2) used a cognitive neuroscience method – eye tracking, 3) conducted a survey with closed-ended questions, and 4) interviews for the in-lab study. While eye-tracking allows us to investigate people's perceptions of different scenarios, where automated technologies make decisions, the questionnaire enabled us to explore socio-technical imaginaries of people even further. Moreover, during in-lab data collection, in-depth introductory and follow-up interviews were conducted, where participants were also asked to reflect on the experiment and the survey. The reason for conducting in-lab interviews was to gain more insights into participants' experiences, perceptions, and thought processes during the study, whereas online interviews were not feasible due to the large number of participants. Interviews gave us an additional opportunity to ensure the reliability of the developed study instruments through cognitive testing of the survey instruments and ensure that the respondents' understanding of the survey instruments corresponded with the researchers' ideas.

The survey questions were formulated based on validated instruments and frameworks from prior studies. For example, some questions regarding values toward algorithmic solutions were drawn from previously validated studies, e.g., [50]. Additionally, we conducted qualitative studies in earlier phases of our research, e.g., [29], and relied on existing frameworks to formulate formalized questions in this survey. In this study, we used some socio-demographic variables from the survey, such as participants' country, education level, readiness to use AVs, and prior participation in similar studies, as part of our analysis. These variables were collected to contextualize viewing behavior. However, not all variables collected through the survey were analyzed in this study, as the primary focus is methodological. This article primarily focuses on analyzing viewing behavior, comparing two methods – online and in-lab eye-tracking – to address gaps in prior

research on sociotechnical imaginaries of automated technologies. Therefore, due to the space limitations of this article, the more in-depth analysis of associations between viewing behavior and background variables will remain the focus of future research.

In-lab experiment was conducted within a physical laboratory environment, allowing researchers the advantage of monitoring the experiment, participant interactions, and data collection procedures. The online experiment utilizes a digital platform and the internet to engage with a wider and often more diverse pool of participants. However, the researchers have limited control over the experimental processes and the behavior of participants. Before running the in-lab study, we made test trials that helped refine the technical setup of the eye-tracking system and ensured the validity of the experimental design, allowing for a smoother execution of the main study. Similar test trials were also conducted for the online study to ensure the clarity of scenarios, the survey questions, and the reliability of data collection in a remote setting.

All procedures performed in studies involving human participants were in accordance with the ethical standards of the institutional and/or national research committee and with the European data protection regulations. The study was approved by the Human Research Ethics Committee of the Health Development Institute in Estonia (reference number: 1063). Written informed consent was obtained from all individual participants included in the studies – signed paper-based consent was collected from in-lab participants, and online consent from online participants.

### 3.1. Samples' structure

To design the sample for both online and in-lab experiment, we have used the combination of strategic [51] and representative sampling principles [52]. Besides, we also took into consideration the necessary pragmatic choices [28] usually faced in the online panel and platform studies, such as limited access to panels from different cities, and no monitoring and controlling over the experiment from the researchers' side. The pragmatic approach was also additionally used in designing the sample in the in-lab study, where the purposeful sample was combined with the convenience sample elements and experimental approach with interviewing to ensure the study of the perception of algorithms as a rather abstract topic (see, e.g., [29]).

A purposeful sampling strategy [51] was used for the in-lab experiment (N = 16). Based on this sampling approach, the sample is homogeneous in regard to their place of residence – Estonia, and therefore having similar experiences with experimenting with the autonomous vehicles in the public space. Besides, the study participants were assumingly aware of using algorithms in designing transport solutions, based on their higher education level. However, their perception towards social diversity and automated technologies might vary. Besides, we purposefully planned the sample to be internally heterogeneous to ensure the expression of various views and understandings. So, the small-scale in-lab experiment was not intended for broad statistical generalization, but rather to develop, refine, and test the developed method prior to conducting our large-scale online eye-tracking study. It was used as a methodological cross-check to verify that the online eye-tracking yielded fixation and reaction-time metrics comparable to those of the in-lab eye tracker. However, when scaled up, the in-lab experiment combined with pre- and post-experiment interviews could serve as an independent study in the future. So, the aim of the in-lab study was to technically and methodologically compare the two eye-tracking approaches.

By employing a purposeful sampling strategy [51], we ensured a diverse range of perspectives, allowing for the identification of potential biases in the research design and the opportunity to address them for our online experiment. Therefore, following the principles of purposeful sampling, the sample consisted of a relatively homogeneous group of students as young experts who were potentially aware of algorithm- and AI-based approaches in transport planning. In addition, to diversify the responses, half of the study participants were social scientists (N = 8), and the other half computer scientists (N = 8). Our in-lab participants included engineers and scientists from the field of Computer Science, many of whom have hands-on experience with autonomous vehicles and algorithms. While other participants have been actively involved in studying and evaluating algorithms, providing further depth to their insights. Their technical expertise in relevant areas

allowed us to capture valuable perspectives on both the development and testing processes of autonomous vehicles, which enriched the outcomes of our study. At the same time, the inclusion of Social Scientists was crucial, as they brought essential perspectives on the societal, ethical, and behavioral aspects of autonomous vehicle adoption, ensuring a more comprehensive understanding of the broader implications of the technology.

The online experiment was conducted through RealEye – a web-based eye-tracking platform. The participants were recruited through a crowd-sourced platform, CINT, which includes online panel members from 130 global countries. In this study, we selected participants based on national census data to ensure the territorial representativeness of the study results. Besides, the experiences with diverse transport modes were used as an additional criterion for participant selection. Our large-scale eye-tracking study involves participants from 26 cities around the world (N = 1276). Cities selected for the experiment are Amsterdam (Netherlands), Ankara (Turkey), Berlin (Germany), Buenos Aires (Argentina), Curitiba (Brazil), Guayaquil (Ecuador), Helsinki (Finland), Ho Chi Minh City (Vietnam), Hong Kong (China – special administrative region), Jakarta (Indonesia), Johannesburg (South Africa), London (United Kingdom), Los Angeles (United States), Manila (Philippines), Mexico City (Mexico), Mumbai (India), New Delhi (India), New York (United States), Paris (France), Reykjavík (Iceland), Salvador (Brazil), São Paulo (Brazil), Singapore (Singapore), Stuttgart (Germany), Tokyo (Japan), and Zurich (Switzerland). The experiment was conducted in English. Given that the participants are rather digitally aware and literate, along with the global spread of the CINT platform, where panel members are often engaged in international studies conducted in English, we believe that language did not pose any significant barriers.

In our online experiments, we ensured diversity and representativeness by carefully selecting participants from cities with varying socio-demographic profiles, including those that have hosted autonomous vehicle pilot programs. Specifically, we selected cities based on the Bloomberg Aspen Initiative on Cities and Autonomous Vehicles 2017 [53] list to target cities with exposure to AV technologies, and the CIMI Index 2020 [54], which measures the advancement of the cities in regard to implementing innovative technologies in developing the cities. We included in the study cities with diverse values of the CIMI index, including those with high, middle, and low values. We assumed that individuals from the cities with diverse experiences with smart city technologies might have different experiences with autonomous vehicles as one of the recent innovations, and therefore might have various perceptions of the potential consequences of these solutions. Our initial aim was to recruit participants from 20 cities, with 100 participants from each city (see Appendix 1). However, due to availability constraints and recruitment challenges on the CINT platform, some cities – particularly cities with high CIMI ranks and those listed in the Bloomberg Aspen AV Initiative Cities, which were underperforming – were substituted with other cities from a similar category to maintain the integrity of the sample structure. As a result, we expanded our selection to include an additional 6 cities. The final sample structures of the online and in-lab studies are presented in Table 1.

To further enhance the representativeness of our sample, we utilized the function offered by the CINT platform, which enabled us to draw participants in proportions reflective of the national census. This ensured that our online sample was representative across key demographic variables such as age, gender, and education level. Nevertheless, territorial representativeness within a county remains elusive in online crowd-sourced studies [55] compared to face-to-face surveys [56], and some regions, particularly rural, may still be under-represented. By leveraging the platform's capabilities, we were able to ensure a broad and diverse participant pool, providing a strong foundation for analyzing the cognitive engagement and perceptions of autonomous vehicles across different geographical regions.

## 3.2. Scenario design and stimuli

To study the sociotechnical imaginaries of autonomous vehicles through implementing an eye-tracking approach, we employed a visual narrative scenario method as a stimulus for both in-lab and online experiments [57,58]. We assumed, and as the trial studies proved, the approach enables addressing the uncertainties and complexities of automated technologies in long-term urban mobility challenges and measuring the people's cognitive engagement with the potential consequences of these technologies.

 

**Table 1. Online and in-lab final sample structure.**

| Category | Sub-category | In-lab: No. of participants | Online: No. of participants |
|---|---|---|---|
| Sex | Female | 8 | 522 |
| | Male | 8 | 754 |
| Age | 18-20 | * | 58 |
| | 21-30 | 11 | 365 |
| | 31-40 | 3 | 367 |
| | 41-50 | 2 | 293 |
| | 51-60 | * | 148 |
| | 61+ | * | 94 |
| | Not known | * | 5 |
| Education** | Basic | * | 54 |
| | Secondary | * | 229 |
| | Higher | 16 | 977 |
| Readiness to use a fully automated AVs | Yes | 8 (50%) | 833 (65.3%) |
| | No | 4 (25%) | 345 (27%) |
| | Don't know, can't say | 4 (25%) | 98 (7.7%) |

*Category not included

** **Basic education** *includes early childhood education, primary education, and lower secondary education.* **Secondary** *includes upper secondary education and post-secondary non-tertiary education.* **Higher** *includes short-cycle tertiary education, bachelor's degrees, master's degrees, and doctoral degrees. Note: We followed international studies like the European Social Survey for education categories. However, to ensure efficiency, we merged the education categories due to variations in the RealEye platform and CINT across cities.*

We used autonomous vehicle as an empirical case study and considered it as a proxy to study the data-based public services offered in the cities, which are assumed to provide equal access to transport and city infrastructure, such as hospitals to all residents despite their social background. Our experiment relies on a previous study by Awad et al. [9] who designed the Moral Machine to explore the moral dilemma of autonomous driving. However, our study strived to develop further this idea to explore sociotechnical imaginaries on AVs and perceptions of social diversity to address a critical gap in the literature: how automated systems account for diverse social identities. Understanding perceptions of diversity is crucial, as these perceptions influence trust, fairness, and the inclusivity of algorithmic systems. In our experiments, we used scenarios to explore one of the ethical concerns related to machine learning, namely, how people perceive social diversity when an autonomous public bus or a driver has to make a decision in case the bus experiences a sudden brake failure.

To design the scenarios, we followed Crenshaw [59] and Vertovec [60] approaches that diversity shall not be analyzed only in terms of ethnicities, race, and countries of origin, but rather in terms of the intersectionality that affects where, how, and with whom people live — super-diversity. It is argued by Crenshaw [59] that theorists need to take both sex and race on board and show how they interact to shape the multiple dimensions of people's experiences. To avoid the one-dimensional appreciation of contemporary diversity, we depicted passengers, pedestrians, and the driver by using the intersection of sex and race with other social categories that are dominant in our society and affect and shape how people interact with AVs.

Two sets of scenarios were shown to participants in a randomized sequence within each set, and across the two sets, for both studies. Randomization was used to eliminate the selection bias and how confounders potentially impact our results. One set of scenarios was about the autonomous public bus that needs to decide between saving pedestrians in the street or passengers in the bus, so the responsibility relies on the algorithms to make such a decision. While in the other set of scenarios, the only difference was that the responsibility relies on a human being (driver). The scenario

categories used in this study to depict social demographic background of pedestrians and passengers are the following: nationality (EU/Asia/US/Africa), income (rich/homeless), micro-mobility (scooter/robot delivery), age (young/adult/old), disability (wheelchaired), COVID-19 (vaccinated/not vaccinated), residency (urban/rural). Some of the categories in this study were visually distinguishable, while others, such as COVID-19 status, were represented using symbols. However, for each scenario, participants had the option to click on 'Show Description' for additional details. We first conducted the in-lab experiment (experiment running 27 April-13 May 2022) to test the developed method and collect small-scale data, and based on that, slightly redesigned the online experiment (experiment running from 13 September-28 October 2022). Minor differences in designing the scenarios in the online experiment were needed, due to the time limit (see Table 2).

The scenarios are designed to compare how people perceive social diversity when an autonomous bus or a regular bus faces a life-or-death situation. Participants were asked, "Which group of people *will* be saved first?" We used *will* instead of *should* as proposed by Kastner [44] since we were particularly interested in people's imaginaries in regard to automated decision-making – what they think that the autonomous bus or the driver will choose to save: two passengers in the bus or two pedestrians crossing the road – instead of what should be the case which implies the moral or normative decision. Also, we used the word *save* instead of *kill* following Kastner's [44] suggestion, since saving the group would be the morally correct choice and reduce the anxiety over choosing to kill. Moreover, we tried to limit participants' moral reasoning (such as the trolley problem, respecting road signs) as much as possible because we were particularly interested in analyzing participants' attention to social diversity, limiting other moral reasoning. Nevertheless, saving someone over someone inherently encompasses moral and ethical considerations.

In the experiment, participants were briefed on the procedure and scenarios. In-lab, they could ask researchers for further guidance, unlike in the online version. However, the online experiment included a section for feedback. In both formats, a two-second white screen buffer was used between slides for eye relaxation and focus. Participants began by clicking "start", then on a split screen, they chose between two options by clicking an image: save two pedestrians (right) or two passengers (left).

### 3.3. In-lab and online eye-tracking experiments

In-lab eye-tracking experiment occurred in an approximately 30m$^2$ room where participants sat at the desk with the head stabilized at the chin rest located in the middle of the monitor and 42–62 cm away from the screen that showed the scenarios in the form of images. The eye-tracking device was positioned on the desk below the screen. Whereas the online experiment was conducted using RealEye's platform, which predicts a person's gaze point using only webcams as data collectors. People could take part in the online experiment from their home at a convenient time using a laptop or desktop PC, thus increasing the chances for higher data acquisition [61]. Since the experiment was unsupervised, RealEye used algorithms to assess whether people's head is centered on the screen and their eyes are properly situated on the screen.

**Table 2. Experiment design: in-lab vs. online.**

| Aspect | In-lab Experiment | Online Experiment |
|---|---|---|
| Scenario Set | 2 sets: one about AV and the other about non-AV | 2 sets: one about AV and the other about non-AV |
| Scenario Category | 7 categories * | 7 categories * |
| Scenarios Presented | 2 scenarios/category (14 scenarios/set) | 1 scenario/category (7 scenarios/set)** |
| Timeframe | April-May 2022 | September-November 2022 |
| Experiment Duration | No time constraint | Subject to max. 30 min |

* The scenario categories utilized were the same across both in-lab and online experiments.

** Due to the time limit for the online experiment, we have selected only 7 scenarios for each set

For the in-lab experiment, we used an SR Research Portable Duo eye-tracking device, which produced stable, low-noise, binocular recordings up to 2000 Hz with the head stabilized and an average accuracy down to 0.15°. The blink recovery was around 0.5 milliseconds (ms) with a 2000 Hz sampling rate and was also compatible for participants wearing glasses or lenses. The eye-tracker was focused on the glint in the participants' eyes, followed by a preliminary 9-point calibration of the device. Since the online experiment was webcam eye-tracking, data were captured at a sampling rate varying between 30 Hz and 60 Hz, depending on the participants' webcam and internet connection speed.

RealEye uses the computing power of a regular PC/laptop to run AI that analyzes images coming from a webcam. The AI detects a participant's pupils and predicts a gaze point with an accuracy of ~110px (~ 1.5 cm) and an average viewing angle error of ~4.17 degrees. This allows analyzing users' interaction on a website with precision reaching the size of a single button. The platform did not record any image or sound to the servers, only storing the gaze point predictions in the form of basic text data similar to "Timestamp: 10, GazePointX: 200, GazePointY: 330". RealEye and CINT are fully compliant with GDPR and guarantee data protection for each participant. The camera was switched off immediately after each test.

### 3.4. Data analysis

We analyzed and compared eye-tracking data based on the average fixation duration, cumulative (total) fixation duration, and reaction time of participants, as expressed to the visual stimulus and scenarios presented. These metrics serve as proxies for cognitive processing and attention. Average fixation duration is the average length of time a participant's eyes stop scanning the scenarios and fixate the vision in one point/place. Cumulative fixation duration is the sum of all fixation points (counts); it indicates the overall time spent fixating within the given stimuli. Reaction time is the time from when the visual stimuli appeared to when an action (i.e., selecting a picture) was taken (for the online experiment, there was a limited exposure time to each scenario).

Specifically, the fixation duration provides insights into the depth of cognitive engagement, while the reaction time sheds light on the decision-making time of participants. In general, longer fixations are associated with deeper cognitive processing and attention, while shorter reaction times indicate quicker decision-making or recognition processes. Therefore, the differences in these metrics between the two methods can provide valuable insights into how participants perceive AV and non-AV scenarios.

## 4. Results

### 4.1. Differences in cognitive engagement across in-lab and online eye-tracking methods

We have compared the cognitive engagement of people when observing autonomous vehicles versus non-autonomous vehicles. We assumed that there is a difference in how people perceive autonomous vehicles versus non-autonomous vehicles, and this is due to many factors such as contextual familiarity, personal relevance and moral deliberation, social desirability, and external influences.

Cognitive engagement refers to the depth and intensity with which participants process information. In eye-tracking studies, fixations serve as a proxy for cognitive engagement, with longer fixation duration indicating greater cognitive processing and attention paid to specific aspects of the presented stimuli. The cognitive intricacies tied to AVs mirror collective imaginaries of a technology-driven landscape. In this study, two sessions of scenarios were almost identical, with the only difference being the type of vehicle. We were interested in seeing whether there was a difference in how people perceive autonomous vehicles in a life-death situation compared to manual vehicles. It is important to highlight that there are some differences in the experiment design and the sample between the online and in-lab study (see Table 2).

Online and in-lab eye-tracking findings revealed a tendency for higher fixation during scenarios involving algorithmic decisions in the case of AVs. Specifically, in the online study, the average fixation duration in AV scenarios was about

5.8% more compared to non-AV scenarios. In the in-lab experiment, the average fixation duration of AV scenarios was slightly higher than non-AV scenarios (about 1.71% more compared to non-AV scenarios). The longer fixation duration (see Table 3–4) indicates that people were processing more information and thinking more deeply about the decision the algorithm, or the driver, was making.

The contrast between the online and in-lab eye-tracking experiments offers intriguing insights into how participants perceive and interact with AV and non-AV scenarios. In the online experiment (Table 3), all scenarios—ranging from "Nationality" to "Urban/Rural"—displayed significant differences (p < 0.001) in fixation durations between AV and non-AV settings. This denotes a clear distinction in how participants engaged with the two types of scenarios. Specifically, they exhibited longer fixation durations for AV scenarios, indicating deeper scrutiny or increased processing when considering autonomous vehicles.

Conversely, the in-lab experiment data (Table 4) painted a different picture. The comparative analysis of fixation durations across distinct scenario categories in the in-lab experiment, despite revealing observable disparities between AV and non-AV visual stimuli, the small sample size did not allow to make strong conclusions. The in-lab experiment engaged in purposeful sampling, curating a participant pool with specific characteristics and presumable predispositions towards the stimuli, which could potentially moderate the perceptual variances across scenarios. This requires delving deeper into potential participant-specific factors that might elucidate these perceptual tendencies.

**Table 3. Fixation duration per category from the online experiment (average, st. deviation σ), N = 1276.**

| Scenario Categories | AV | | | non-AV | | | T-test |
|---|---|---|---|---|---|---|---|
| | Fix. (ms) | σ | Cumulative Fix. (s) | Fix. (ms) | σ | Cumulative Fix. (s) | |
| Nationality | 253 | 55.81 | 7.00 | 237 | 55.47 | 6.80 | 10.53 |
| Income | 251 | 56.79 | 7.68 | 238 | 56.92 | 6.92 | 8.74 |
| Micro mobility | 251 | 56.40 | 7.00 | 235 | 54.82 | 6.77 | 10.66 |
| Age | 250 | 56.32 | 6.67 | 237 | 55.70 | 6.27 | 8.48 |
| Disability | 250 | 57.14 | 6.46 | 235 | 56.78 | 6.25 | 9.69 |
| COVID-19 | 252 | 56.51 | 7.03 | 237 | 57.01 | 6.63 | 9.69 |
| Urban/Rural | 251 | 55.45 | 7.26 | 239 | 56.64 | 6.74 | 8.32 |

*Each scenario category showed statistically significant differences (p < 0.001) in fixation duration comparing AV and non-AV*

**Table 4. Fixation duration per category from the in-lab experiment (average, st. deviation σ), N = 16.**

| Scenario Categories | AV | | | non-AV | | |
|---|---|---|---|---|---|---|
| | Fix. (ms) | σ | Cumulative Fix. (s) | Fix. (ms) | σ | Cumulative Fix. (s) |
| Nationality | 222 | 42.83 | 14.68 | 213 | 41.63 | 13.41 |
| Income | 202 | 45.90 | 12.91 | 217 | 26.32 | 15.22 |
| Micro mobility | 213 | 25.41 | 9.60 | 218 | 31.26 | 14.26 |
| Age | 227 | 44.24 | 11.91 | 210 | 24.39 | 13.01 |
| Disability | 214 | 49.23 | 15.29 | 220 | 36.27 | 19.94 |
| COVID-19 | 232 | 45.02 | 15.30 | 216 | 26.69 | 19.85 |
| Urban/Rural | 232 | 41.08 | 15.36 | 222 | 27.38 | 25.55 |

*Each scenario category did not show statistically significant differences in fixation duration comparing AV and non-AV due to the relatively small sample size (N = 16)*

## 4.2. Exploring cognitive engagement: Viewing behaviors within in-lab and online eye-tracking methods

There are several aspects within the method that might explain the observed differences in the results across the two methods. Viewing behaviors were analyzed within the in-lab eye-tracking study and the online experiment. A particularly intriguing result emerged in the viewing behaviors between the Data&Algorithm-Informed (DAI) group which is potentially more informed about data and algorithms, and the Data&Algorithm-Uninformed (DAU) who may lack technical knowledge and experience with data and algorithms.

In both experiments, DAU and DAI demonstrated a higher average fixation duration toward AV scenarios (Fig 1). However, the cumulative fixation duration provided a different picture in-lab – DAI and DAU tend to fixate longer towards non-AV.

As we dig deeper into the specifics in the sections below, the intricacies of these differences and their implications for societal imaginaries concerning autonomous vehicles become more evident.

### 4.2.1. Fixation duration variations within online eye-tracking.
The study's analysis of online viewing behaviors provides insights into how different groups of people perceive and react to autonomous vehicle scenarios. The findings suggest that DAI, who are aware of data and algorithms, tend to have shorter average fixation duration and take less time to make decisions compared to DAU (Fig 1A). Additionally, both groups showed a tendency to fixate longer on AV than on non-AV.

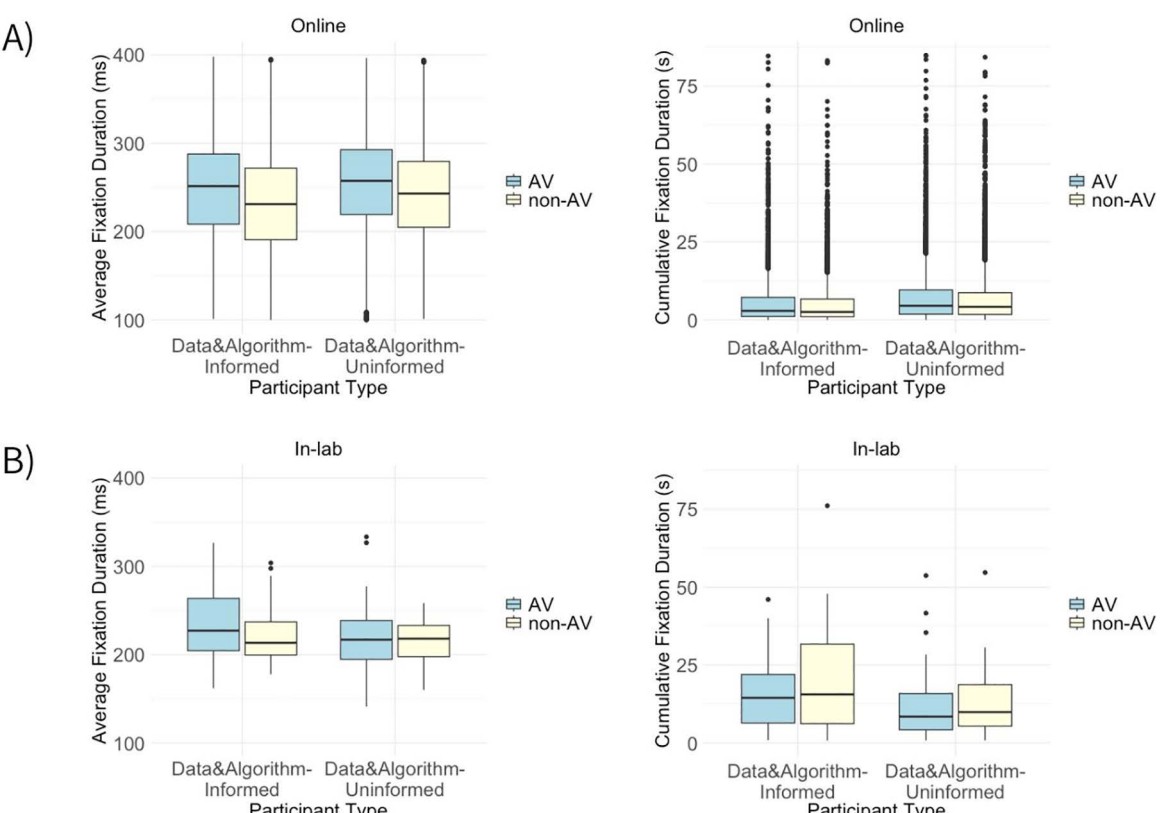

**Fig 1. Average and cumulative fixation duration differences between DAI and DAU.** Average fixation duration (left boxplot) and cumulative fixation duration (right boxplot) difference between DAI and DAU in online experiment (A), and in-lab experiment (B).

Online viewing behaviors were analyzed between and within 1) Data&Algorithm-Informed (n = 508) – people who are aware of data and algorithms and *participated* at least once in similar studies, and 2) Data&Algorithm-Uninformed (n = 768) – those who *never participated* before in similar studies and assumingly might not be aware of the data and algorithms. In addition, viewing behaviors were analyzed across two sets of scenarios – *autonomous vehicle* and n*on-autonomous vehicle*. For DAI participants, classified as such due to their prior participation in similar studies and presumed familiarity with algorithmic systems, the average fixation duration (the depth of fixating on one point) and total fixation duration showed a trend towards being higher in AV scenarios. However, the t-test revealed that this difference was not statistically significant (t(7080) = 1.95, p = 0.051). For DAU participants, total fixation duration was statistically significantly longer in AV scenarios compared to non-AV scenarios, as indicated by the t-test results (t(10748) = 2.11, p = 0.034).

To examine the effects of group (DAI vs. DAU) and scenario setting (AV vs. non-AV) on fixation time, a Two-Way ANOVA (Table 5) was conducted because it (i) allowed simultaneous estimation of two main effects plus their interaction and (ii) is known to be robust when data sizes are large and reasonably balanced [62]. The analysis revealed a significant main effect of group, $F(1, 17860) = 171.26$, $p < 0.001$, partial $\eta^2 = 0.009$, indicating that fixation times were significantly different between the DAI and DAU groups. There was also a significant main effect of scenario setting, $F(1, 17860) = 8.11$, $p = 0.004$, partial $\eta^2 = 0.0005$, suggesting a significant difference in fixation times between AV and non-AV scenarios. However, the interaction between the groups and scenario setting was not significant, $F(1, 17860) = 0.00085$, $p = 0.976$, indicating that the effect of the scenarios on fixation time did not differ between the DAI and DAU groups.

The assumptions of normality and homogeneity of variances were assessed using Shapiro-Wilk and Levene's tests. Shapiro-Wilk on a random 5,000-residual subset (upper practical limit for the test [63]) indicated right-tail skew, $W = 0.663$, $p < 0.001$, likely due to the inherent skewness of the data derived from 26 cities. Levene's test indicated unequal variances across the four experimental cells (DAI-AV, DAI-non-AV, DAU-AV, and DAU-non-AV), $F(3, 17860) = 24.72$, $p < 0.001$. To guard against bias from these violations, we replicated the analysis using (a) a Welch heteroscedastic two-way ANOVA (HC3 estimator) and (b) a generalized linear model with a Gamma error distribution and log link – both appropriate for positively skewed duration data. The Welch test revealed group $F = 182.1$, $p < 0.001$; scenario $F = 8.25$, $p = 0.004$; interaction $F = 0.00091$, $p = 0.98$. The Gamma-log GLM showed +31% for DAU relative to the reference category DAI for AV scenarios ($p < 0.001$), −6% for non-AV relative to AV within DAI group ($p = 0.036$), and no statistically significant interaction ($p = 0.71$). Thus, both tests replicate the group and scenario main effects and the null interaction observed in the classical ANOVA, with overlapping 95% confidence intervals.

To validate the findings, a Wilcoxon Signed-Rank test was conducted to compare fixation times between AV and non-AV settings within each group. Wilcoxon Signed-Rank tests conducted at the fixation-event level showed a significant AV–non-AV difference for DAI ($p = 0.029$) and a highly significant difference for DAU ($p = 0.0001$). When the test is run on participant-level means, the DAU effect remains significant ($p = 0.044$) while the DAI effect becomes a non-significant trend ($p = 0.105$); thus, the overall scenario effect reported in the ANOVA is robust, although its strength within the expert group is modest.

**Table 5. Results of two-way ANOVA for fixation time comparing DAI vs. DAU groups and AV vs. non-AV Setting*.**

| Source of Variation | Sum of Squares | Degree of Freedom | F value | P value |
|---|---|---|---|---|
| Setting: AV vs non-AV | 678368644.95 | 1.0 | 8.11 | 0.004 |
| Group: DAI vs DAU | 14328778579.25 | 1.0 | 171.26 | <0.001 |
| Setting x Group | 71910.79 | 1.0 | 0.00085 | 0.976 |
| Residual | 1494251690557.35 | 17860 | | |

*DAI = Data&Algorithm-Informed; DAU = Data&Algorithm-Uninformed; AV = Autonomous Vehicle; non-AV = not Autonomous Vehicle (manual vehicle)

Furthermore, an interesting observation was the heavy-tailed boxplot (Fig 1A, left image) and the presence of outliers with notably prolonged fixation durations (Fig 1A, right image). A significant factor to consider is the context [64], such as the geographical diversity of our participant sample, with individuals coming from various cities, each carrying its unique set of cultural, socio-economic, and educational backgrounds. Such diversity can explain variations in how participants process, interpret, and engage with the stimuli. For instance, visual content that might be easily understood in one city could be novel or require deeper contemplation for participants from another. The association between outliers in fixation times and participants' countries was assessed using a chi-square test. which test yielded $\chi^2 = 1284.06$, df = 60 and $p < 0.001$. The low p-value suggests a statistically significant association, indicating that the distribution of fixation times varies significantly across different countries.

Fig 2 represents heatmaps aggregating fixation data from DAI and DAU participants in a specific scenario and is intended as an illustrative example to show the differences in attention patterns between the two groups for AV and non-AV scenarios (to see the difference between DAI and DAU by means of average fixation time across all scenarios, see Fig 1). The heatmaps' warmer colors indicate regions of intensified fixations. DAI, with their overall shorter fixation durations than DAU, appears to distribute their attention more in specific areas across the scenarios. This suggests an analytical perspective on some elements of stimuli, possibly due to their data and algorithmic expertise. Conversely, DAU,

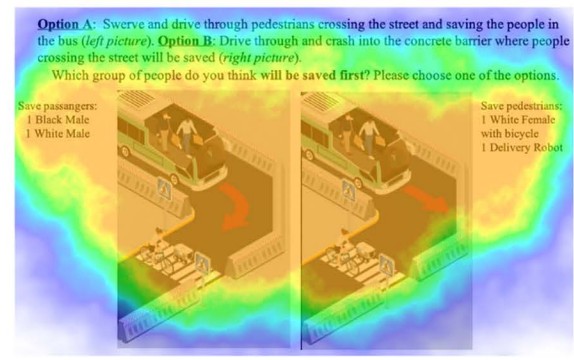
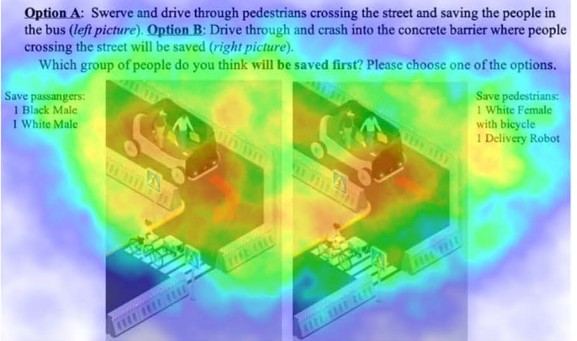
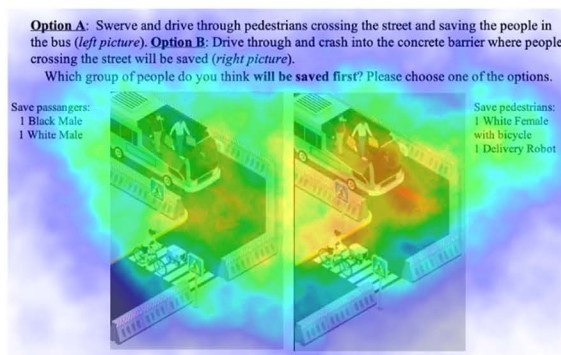

**Fig 2. Online eye-tracking heatmaps of DAI and DU.** Warmer colors (e.g., red and orange) indicate areas with higher attention or fixation durations, and cooler colors (e.g., green and blue) represent areas with lower attention.

with their longer fixation durations, seem to focus more intently on AV scenarios, possibly indicating areas of intrigue or uncertainty.

**4.2.2. Fixation duration variations within in-lab eye-tracking.** Similarly to online results, the in-lab experiment showed that both DAU and DAI fixated longer on a single point in the AV scenarios. However, unlike the online experiment, the cumulative fixation duration in-lab showed different results – the overall time spent fixating on the given stimuli on average was higher in non-AV (Fig 1B). Specifically, DAI participants (background in computer science) demonstrated higher total fixation duration in non-AV in comparison with DAI (background in social science), who showed slightly higher total fixation duration in non-AV than AV.

The heatmaps (Fig 3) represent aggregated data across all scenarios presented in the experiment. However, the chosen image is only an illustrative one to provide an overview of the differences in gaze patterns between the DAI and DAU groups across the two sets of experimental conditions. The heatmaps make it clear that their attention was drawn significantly to the "additional information" in the non-AV scenarios provided alongside the images, which described the passengers and pedestrians. This behavior indicates a shared interest in understanding the human context and social dimensions of the scenarios. However, how they engaged with the whole content showcased differences in their analytical approaches.

Autonomous scenarios Non-autonomous scenarios

Data&Algorithm-Uninformed

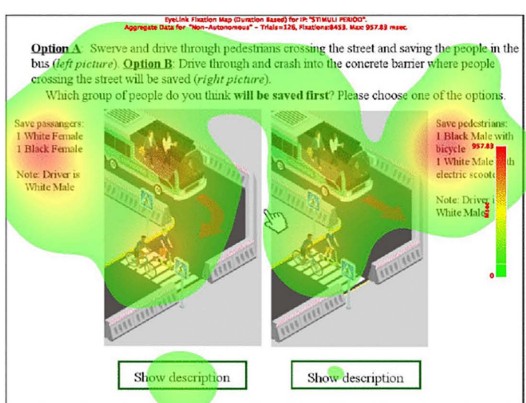

Data&Algorithm-Informed

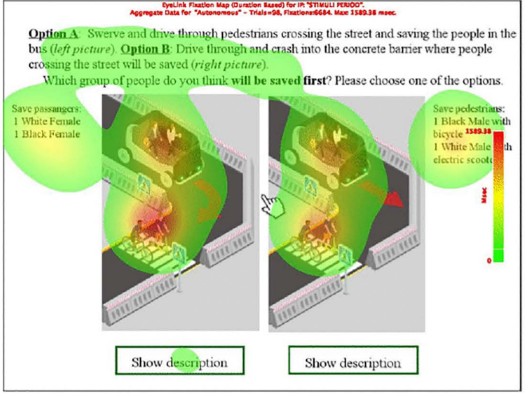
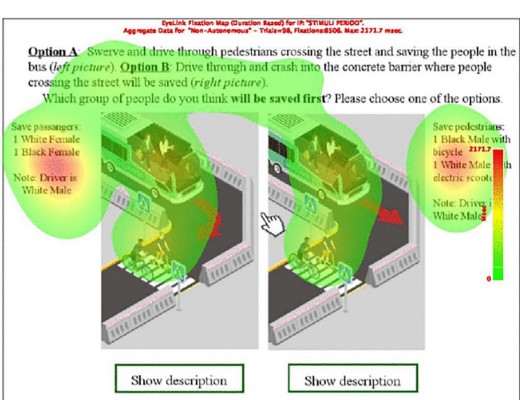

**Fig 3. In-lab eye-tracking heatmaps of DAI and DU.**

On one hand, DAU participants engaged more holistically with the scenarios. Their gaze was spread evenly, oscillating between the visual stimuli and the supplementary text, suggesting an integrated approach to understanding both the technological and social facets of the scenarios. This was also reflected during the interviews, where DAU intended to not only understand but also position AVs within a broader societal and ethical frame, and critically evaluated biases in human-driven vehicles where the social demographics of the driver have been depicted in non-AV scenarios.

On the other hand, DAI participants, with their technical background, demonstrated a more segmented approach. In AV scenarios, their primary focus was on the images themselves, possibly trying to discern technological or mechanical attributes. In contrast, for non-AV scenarios, their attention shifted predominantly to the contextual text. This implies that while they were keen on understanding the technological nuances in AV scenarios, they leaned towards grasping the socio-contextual narrative in non-AV situations. Based on the interview data, DAI showed a critical evaluation of AV scenarios, as they noted that the mere presence of diverse data in algorithmic systems does not automatically guarantee equitable or fair outcomes in AV decision-making processes. They argue that while diverse data is a critical element in developing robust and inclusive algorithms, how this data is processed, weighted, and applied within the algorithm is equally important. However, longer fixation duration in non-AV scenarios shows their attempt to critically evaluate, or perhaps even challenge, the constructs and decisions embedded within human-driven vehicle representations. According to them, humans share their own values and beliefs, and therefore, this might lead to biased decisions, whereas the algorithm decides based on the type and availability of data – detached from human prejudice and biases.

Determining the precise differences behind such attention distribution requires more granular analysis in further studies. This involves breaking down the scenarios into specific Areas of Interest (AOI) and examining cognitive engagements within these AOI.

**4.2.3. Exploring decision-making: Reaction time within in-lab and online eye-tracking.** Reaction time, when evaluated alongside fixation durations, offers a deeper understanding of participants' decision-making processes during the experiment. The reaction time is the period from the presentation of the stimuli to the moment of the decision made by the participants. The analysis of reaction time complements our understanding of the fixation duration variations observed in the previous sections, offering valuable insights into how these variations are indicative of participants' cognitive engagement and internal deliberations. Faster reactions could indicate instinctual responses, possibly influenced by prevalent societal narratives or personal beliefs about AVs. Conversely, extended reaction times might suggest participants grappling with their internal biases, beliefs, or the contemplation of potential outcomes in the presented scenario.

The reaction time data (Table 6) showed that in-lab DAI and DAU average reaction time was longer in non-AV, whereas online DAI and DAU showed slightly longer reaction time in AV scenarios. In addition, when comparing the differences of in-lab DAI versus DAU, the average reaction time for both AV and non-AV scenarios is longer for DAI – they took more time to make a decision than DAU. This confirms the findings shown above that overall DAI fixated longer than DAU. Nevertheless, the online DAI and DAU average reaction time was almost identical.

Table 6. Average reaction time in seconds comparison between DAI and DAU in online and in-lab experiment.

| Scenario Set | In-lab experiment | | Online experiment | |
|---|---|---|---|---|
| | DAI* | DAU** | DAI* | DAU** |
| AV | 19.10 | 15.89 | 8.14 | 10.25 |
| Non-AV | 23.94 | 18.19 | 8.02 | 10.20 |

*DAI=Data&Algorithm-Informed **DAU=Data&Algorithm-Uninformed

In addition, in-lab participants were people who showed particular interest in joining the experiment, and especially a group of participants who were engaged directly with AV algorithms, which might affect the reaction time because of their interest in the topic.

## 4.3. Evaluation of data quality in online and in-lab eye-tracking experiment

The differences and similarities in the method could be explained by the fact that, due to technical aspects (Table 7), the quality of measurement differs across methods.

The experiment monitoring during the in-lab experiment was much higher in comparison with the online experiment. This explains the difference in data quality between the experiments. The in-lab experiment was supervised by the experimenter, who sat at one end of the desk with a laptop computer that monitored the experiment and calibrated the eye-tracker (if needed) for proper gaze recording. However, the online experiment was unsupervised, so there was no control during the online experiment. During the in-lab experiment, after the first fixation has been accepted, the remaining calibration targets are displayed in sequence, and fixation values are collected for each scenario. The calibration system presents these targets in a random order, which discourages participants from anticipating the location of the next dot and cascading away from the current target before it disappears. However, it was important to remind the participant to look at each calibration dot until the next target appears. After the calibration, the validation process started. In each validation phase, 9 dots appeared one after the other, in random order, on each of the four edge positions of the calibration and in the center.

Validation showed the experimenter the gaze position accuracy achieved by the current calibration model. When the participant fixates these dots, the calibration model is used to estimate the gaze position of the participant, and the error (difference between the actual target position and the computed gaze position) is calculated. If a position was not successfully validated, the participant was sent back to the instructional page and had to recalibrate. If the eye-tracker was recording accurately, then the experimenter pressed the key to continue the experiment, and in case there were deviations, then the calibration was performed until the eye-tracker was showing high accuracy of eye movements. To ensure that the accuracy of the calibration parameters is maintained during the whole experiment time, a drift check was performed every $5^{th}$ scenario by showing a dot in the center to reorient the participants' eyes to the center of the screen. This resulted in acquiring only high-quality data, which was not the case with the online experiment.

In the online settings, the system of calibration and validation was all automated. A standard 40-point calibration on RealEye software was shown to each participant in 3 backgrounds – grey, black, and white- to minimize the influence of the monitor's light intensity during the eye-tracking experiment. The calibration is then validated on 9 points. After successful calibration and validation, participants completed two sets of scenarios shown in random order (same as in-lab)

**Table 7. Technical comparison between in-lab and online eye-tracking.**

|  | In-lab experiment | Online experiment |
|---|---|---|
| **Eye-tracker type** | hardware eye-tracker | webcam eye-tracker |
| **Eye-tracker mode** | Head-fixed | head-free/remote |
| **Sample rate** | 2000 Hz | 30-60 Hz * |
| **Average Accuracy** | $0.25° - 0.5°$ | $4.17°$ |
| **Screen Resolution** | $1920 × 1080$ pixels | 1024x968 pixels or more* |
| **Calibration** | Preliminary 9-point calibration Drift check after $5^{th}$ slide | Preliminary 40-point calibration Automated control of the participant's head position |
| **Sample size** | N = 16 | N = 1276 |

*Depending on the participants' device (desktop PC/Laptop)*

– autonomous scenarios and non-autonomous scenarios. The task was self-paced, meaning that the next scenario started whenever participants clicked on one of the images (left image or right image). Since the experiment was all automated, in case participants were facing any technical issues, such as a poor internet connection or stimulus loading, then the skip URL was sent to participants to end the experiment.

Also, different from the in-lab experiment, there was no manual drift check feature to monitor the accuracy of the webcam eye tracker throughout the experiment. In case participants were moving their head outside the standard centered frame, then an automated pop-up message was shown to participants to fixate their head in the center of the screen. Or, if participants had many head movements, then an automated message was shown on the screen, i.e., "Many head movements," and sent back to calibration. Only after the participant fixated the head and the webcam was able to locate the eyes, then the experiment resumed. Not being able to control the calibration and validation, and the lack of a drift check feature has significantly influenced the data quality. Each result has been assigned one of the data quality grades: Perfect, Very Good, Good, Average, Low, or Very Low. Almost 90% of the in-lab eye-tracking data are high-quality data, while the average of the online eye-tracking data is Good.

The quality of data obtained from an eye-tracking study is significantly influenced by the sampling rate of the eye-tracker. Higher sampling rates allow for more precise capture of eye movements, thus enhancing the overall quality of the data collected. The online attrition rate is 39% from which 2010 granted camera access, but only 1276 participants have completed the test and provided results, whereas all in-lab participants filled in a pre-registration form, and there was no dropout. The online experiment was opened by 13036 participants, but only 10% of them completed it. The exact number of people invited to pre-register for the in-lab experiment is not known as it was sent to the student email list by different department representatives, and no data is saved of those who opened the link.

## 5. Discussion and conclusions

This study aimed to explore the characteristics, advantages, and limitations of both in-lab and online experiments, highlighting how each approach contributes to the scientific understanding of using eye-tracking technology in the social sciences and to better understand the sociotechnical imaginaries towards autonomous decision-making systems. This is a comparative study that reveals the differences, similarities, and applicability of utilizing in-lab and online eye-tracking methods to study sociotechnical imaginaries of AVs. Building on previous studies such as the MM experiment by Awad et al. [9], this study aimed to develop further by introducing an original method that combines interviews, eye-tracking, survey, scenario techniques, and a crowd-sourced platform. Although this study focuses on AVs, the methods employed here can also be applied to investigate sociotechnical imaginaries in other domains, such as educational technologies or healthcare automation systems.

This study revealed that both in-lab and online eye-tracking methods not only enriched our understanding of eye-tracking in different settings but also illustrated its potential as a versatile tool in social science research, particularly in understanding human cognitive engagement with emerging technologies. Building on the notion that sociotechnical imaginaries are the collectively held "dreamscapes of modernity" through which societies visualize desirable technological futures [2], AVs research has long relied on a narrative in which software promises to cure the "driver problem" by removing human error [24]. Our eye-tracking evidence lends empirical granularity to that imaginary. Across both in-lab and online experiments, we observed a tendency for participants to fixate longer on a single point in AV scenarios, indicating an increased level of attention or interest in AVs. This prolonged fixation at one point in AV scenarios could possibly be driven by the novelty, concern, or moral salience of automated technology, whose imaginaries may be shaped by diverse global perspectives and personal experiences with data and algorithms.

However, when analyzing the cumulative fixation duration and reaction times, eye-tracking data revealed different pictures between the two experimental settings. In the in-lab experiment, participants showed a longer cumulative fixation duration and reaction time for non-AV scenarios, reflecting more concentrated imaginaries among strategically chosen

participants. This indicates a deeper cognitive processing when considering human-driven situations, potentially reflecting a focus on the complexities and unpredictability of human decision-making who share their own biases, values, and beliefs. Thus, longer fixation duration towards non-AV could stem from the point of understanding the limitations of human drivers that AVs aim to overcome, and the responsibility assigned to them. Additionally, our findings align with the observations of Li et al. [11] who claimed that people assigned less responsibility to AVs for traffic accidents that were at fault than to a human driver who was at fault. Nevertheless, due to the small sample size of the in-lab study, further studies are needed to generalize in this regard.

Conversely, the online experiment participants demonstrated a longer cumulative fixation duration and reaction time in AV scenarios. This could be indicative of the participants' efforts to comprehend and evaluate the decision-making processes of AVs, aligning with global trends of interest and concern regarding the deployment and implications of automated technologies. High-stakes scenarios typically carry significant moral and ethical consequences, thus increasing the cognitive load. This pattern of increased fixation on AV-related content aligns with the findings of Araujo et al. [10], who demonstrated that people prefer algorithmic decisions with high-stakes impact decisions.

Furthermore, our eye-tracking study revealed significant insights into how participants, categorized as Data&Algorithm-Informed (DAI) and Data&Algorithm-Uninformed (DAU), engaged with AVs. In the online setting, DAU participants, with less technical knowledge about data and algorithms, displayed longer fixation durations on AV scenarios compared to their DAI counterparts. The observed variances in the online experiment suggest that diverse factors such as personal experiences, domain-specific knowledge, socio-demographics [10], cultural and geographic differences [65] between online participants may influence the perception towards algorithmic decision-making. This highlights the need for more granular studies to delve into these differences. While our findings suggest that these differences may play a role in shaping perceptions, we did not specifically analyze the impact of individual factors such as personal experience, demography, or culture, as it was out of our scope in this study.

In contrast, in-lab results showed that DAI participants, more familiar with technical aspects, had a greater overall fixation duration than DAU, with a notable interest in non-AV scenarios. DAI highlighted that algorithms are detached from human biases and prejudices, and they expressed skepticism about the direct relevance of diversity in the context of algorithmic systems. This might explain their heightened attention to non-AV scenarios, as they grappled with understanding how diversity might impact human decisions. However, these results cannot be generalized due to the small sample size, as the in-lab study was used as a validation for the online experiment to gather feedback and varied perspectives on the experiment rather than to achieve representativeness. Nonetheless, the in-lab findings show the need for further research to understand how diverse backgrounds and knowledge levels shape perceptions and cognitive engagement with AVs and algorithmic decision-making.

This paper makes a significant contribution to the interdisciplinary field of cognitive research and technology perception, particularly in understanding public perception towards AVs. Our findings, which showed variations in visual attention between DAI and DAU participants, highlight the complex interplay between technical understanding and social perceptions of AVs. Building upon prior work that conceptualizes sociotechnical imaginaries as collective visions of desirable futures shaped by technological advancements [2], our findings highlight the "society-in-the-loop" challenge identified by Rahwan [66], emphasizing the critical need for public participation in shaping algorithmic behaviors in AVs. Awad et al. [9] further claim that there may be cultural shifts, yet we do not know the specific changes and mechanisms behind those shifts. Before establishing machine behavioral parameters, it is imperative to understand and incorporate diverse societal perceptions, ensuring that the integration of AV technologies aligns with collectively negotiated societal values and contributes positively to social equity and trust [31,66]. Our contribution lies in demonstrating the potential of online crowd-sourced eye-tracking, how this approach could be used in future work to explore the underlying mechanisms and cross-cultural variations in sociotechnical imaginaries. Because automated decision-making employs algorithmic mechanisms that can differ from social phenomena [31], like biases in human behavior and attitudes, we need new methods to study these mechanisms.

Thus, this research advances the methodological field in studying sociotechnical systems by combining crowd-sourced platforms with online eye-tracking technology to scale up research and explore global sociotechnical imaginaries, and provides crucial insights for policymakers, developers, and researchers. It underscores the importance of considering diverse perspectives and knowledge bases in designing and implementing emerging technologies, thereby contributing to a more inclusive and socially informed approach to technological advancement. Earlier online eye-tracking [67,68] and crowd-sourced [49,69] studies demonstrated that large-N gaze data can be captured online, but applied the technique mainly to interface layout or mobile usability questions. By porting these approaches to the study of global sociotechnical imaginaries – and using our in-lab experiment to validate the results – we show that crowd-sourced eye-tracking can empirically assess real-time perceptual engagement with emerging technologies from a diverse sample, thereby offering measurable insights into how these imaginaries are enacted during decision-making processes. Additionally, the in-lab experiment, as a stand-alone study, provides detailed snapshots of participants' sociotechnical imaginaries and enables conducting interviews to unpack individual interpretations and better interpret the gaze data, even though larger, more diverse samples are needed to generalize these findings. In addition, the comparison of in-lab and online eye-tracking methods aims to contribute to existing methodological discussions, where increasing attention is being paid to online (semi-experimental) studies.

Our research highlights the need for universal regulations governing algorithmic solutions, especially considering these technologies that transcend borders and are adopted across various countries. Although several guidelines exist for adopting AI solutions (see, e.g., [70]), their implementation has proven challenging across different social contexts and even with the same solutions used in different locations. In this context, the EU AI Act [71] offers a promising legal framework for promoting responsible AI. However, our findings suggest that its current provisions lack specificity regarding how social diversity and contextual variations should be systematically integrated into algorithmic systems. This gap is significant, as prior research on sociotechnical imaginaries [2,72] highlights the essential role public perceptions and cultural contexts play in shaping acceptance and trust toward automated systems. Diversity and anti-discrimination have been central themes of the EU AI Act [71]. Our study illustrates how new methods like online eye-tracking techniques compared to in-lab techniques can be effectively utilized to capture variations in the perception and understanding of social categories across diverse contexts, thus informing ongoing debates on the feasibility of universal principles and/or the necessity of context-specific adaptations of regulations such as the EU AI Act. Our empirical findings confirm these insights by demonstrating how diverse knowledge levels (DAI vs DAU) and scenario settings (AV vs non-AV) distinctly shape cognitive engagement and perceptions towards automated decision-making. Such insights call for incorporating broader societal imaginaries into AV design practices, moving beyond purely technical specifications toward socially inclusive design processes.

For AV designers and developers, this implies the necessity of embracing participatory methods and cross-cultural comparative approaches early in the design phase, ensuring technologies reflect and accommodate diverse user perceptions and local values. For policymakers, our results underline the importance of establishing flexible, context-sensitive governance mechanisms that explicitly address sociotechnical disparities and public concerns across diverse geographical and socio-demographic contexts. More broadly, our results highlight that algorithmic solutions are not merely technical artifacts but socio-cultural entities embedded within complex societal systems. Consequently, AV development and deployment must account for macro-level social differences and individual-level cognitive and cultural mechanisms to ensure equitable societal impacts. A closer examination of macro and individual-level mechanisms will be essential for guiding future studies and developing effective policies, fostering socially responsive innovation and responsible technological integration.

## 5.1. Limitations and further studies

We, as authors, are aware of the potential limitations and biases that might arise in this research. While we made every effort to ensure scenarios used in this study were as natural and unbiased as possible, we acknowledge that there might

still be inherent biases that could have influenced participants' viewing behaviors. Therefore, we are aware that visual behavior might be altered since being eye-tracked induces social-norm-based looking behavior [73]. Additionally, the small in-lab sample size did not allow us to generalize the findings, even though the main aim was a technical and methodological comparison of online and in-lab tracking methods. However, we suggest further studies to scale up the in-lab experiment and complement it with other cognitive tools (e.g., electroencephalography (EEG), see [74]) to triangulate and enrich insights into sociotechnical imaginaries. Complementary cognitive tools, alongside eye-tracking, help to capture brain activity, subconscious cognitive and emotional responses to context-sensitive stimuli, which may enable to uncover the hidden mechanisms that shape trust and acceptance and inform socially aligned AV design.

Another limitation was related to the potential sample biases in the online study. While our aim was to get online participants based on their respective city consensus data, it is important to highlight that participants of the online platforms often belong to younger age groups, have higher educational backgrounds, and are more active in digital environments. This sample skewness might lead to an over-representation of views from educated individuals and neglect the perspectives of less educated, and therefore assumingly less aware of the use of algorithmic approaches in planning the transport in the cities. The sample might lead to an underrepresentation of older people, people from rural areas, and vulnerable groups who may have different experiences and viewpoints on such technologies. Moreover, certain cities exhibited a lower response rate compared to others; therefore, we had to add additional cities to maintain the integrity of the sample structure by substituting the cities based on the categories we selected.

Furthermore, we aimed to enhance data accuracy by requiring participants to use laptops rather than mobile phones, as laptops encourage more stationary behavior, improving eye-tracking precision. However, conducting the experiments online without direct supervision made it difficult to control environmental factors like distractions, lighting, and participant focus, which we acknowledge as a potential limitation.

We suggest future studies to dig deeper into specific AOI within the scenarios, and variances across different countries and demographics. Considering the potential influence of culture and country context transportation infrastructure in the cities, knowledge and experiences with technologies, and comparing perceptions across different cities or countries could provide valuable perspectives on how these factors play in shaping the perception towards AVs. Moreover, understanding how individuals with different personal experiences with AVs (e.g., people who use or do not use AVs) engage with specific elements of scenarios, such as images or text can provide richer insights into their reasoning processes and concerns.

## Supporting information

**S1 Appendix. Planned online sample and final online sample.**
(DOCX)

## Acknowledgments

We thank all the respondents for participating in this study. We also thank Walgus OÜ for providing technical support in visually designing the scenarios used in the studies.

## Author contributions

**Conceptualization:** Mergime Ibrahimi, Anu Masso, Mauro Bellone.

**Data curation:** Mergime Ibrahimi.

**Formal analysis:** Mergime Ibrahimi, Anu Masso, Mauro Bellone.

**Funding acquisition:** Anu Masso.

**Investigation:** Mergime Ibrahimi.

**Methodology:** Mergime Ibrahimi, Anu Masso.

**Project administration:** Mergime Ibrahimi.

**Resources:** Anu Masso.

**Software:** Mergime Ibrahimi.

**Supervision:** Anu Masso, Mauro Bellone.

**Validation:** Mergime Ibrahimi, Anu Masso, Mauro Bellone.

**Visualization:** Mergime Ibrahimi, Mauro Bellone.

**Writing – original draft:** Mergime Ibrahimi, Anu Masso, Mauro Bellone.

**Writing – review & editing:** Mergime Ibrahimi, Anu Masso, Mauro Bellone.

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
