## [Decision Letter · Decision Letter 0]

22 Aug 2024

Dear Dr. Ibrahimi,

Thank you for submitting your manuscript to PLOS ONE. After careful consideration, we feel that it has merit but does not fully meet PLOS ONE’s publication criteria as it currently stands. Therefore, we invite you to submit a revised version of the manuscript that addresses the points raised during the review process.

Please try to revise your manuscript and respond to all the reviewers' comments.

We look forward to receiving your revised manuscript.

Kind regards,

Quan Yuan, Ph.D.

Academic Editor

PLOS ONE

“H2020 project Finest Twins (grant No. 856602) and, Development program ASTRA of Tallinn University of Technology for years 2016-2022 (2014-2020.4.01.16-0032).”

3. In the online submission form you indicate that your data is not available for proprietary reasons and have provided a contact point for accessing this data. Please note that your current contact point is a co-author on this manuscript. According to our Data Policy, the contact point must not be an author on the manuscript and must be an institutional contact, ideally not an individual. Please revise your data statement to a non-author institutional point of contact, such as a data access or ethics committee, and send this to us via return email. Please also include contact information for the third party organization, and please include the full citation of where the data can be found.

Reviewers' comments:

Reviewer's Responses to Questions

**Comments to the Author**

1. Is the manuscript technically sound, and do the data support the conclusions?

Reviewer #1: Yes

Reviewer #2: Partly

2. Has the statistical analysis been performed appropriately and rigorously?

Reviewer #1: Yes

Reviewer #2: Yes

3. Have the authors made all data underlying the findings in their manuscript fully available?

Reviewer #1: Yes

Reviewer #2: Yes

4. Is the manuscript presented in an intelligible fashion and written in standard English?

Reviewer #1: Yes

Reviewer #2: Yes

Reviewer #1: This paper focuses on an interesting topic. Based on eye-tracking techniques, the authors investigate sociotechnical imaginaries of autonomous vehicles using both in-lab and online eye-tracking experiments. However, there are several issues to be clarified.

1) Page 4, Line 4. The authors need to review more up-to-date articles in Introduction section. Here are three studies for your information.

[1] A day in the life with an automated vehicle: Empirical analysis of data from an interactive stated activity-travel survey.

[2] Understanding Human Drivers' Trust in Highly Automated Vehicles via Structural Equation Modeling.

[3] Exploring the association between socio‐demographic factors and public acceptance towards fully automated vehicles: Insights from a survey in Australia.

2) Page 8, Lines 145 to 146. The relationship between driver trust on autonomous vehicles and their attention during autonomous driving is also an important factor for traffic safety and public acceptance. See the reference “Latent Hazard Notification for Highly Automated Driving: Expected Safety Benefits and Driver Behavioral Adaptation.”

3) Page 11, Lines 226 to 227. Why not invite automotive engineers/ scientists for the in-lab experiment? As these groups of people are the developer and tester of autonomous vehicles.

4) Page 12, Line 233. The authors should give a description on the list of cities and the corresponding languages.

5) Page 12, Table 1. The authors should report the respondents' experience of using autonomous driving systems.

6) Page 12, Table 1. How do the authors categorize the educational levels?

7) Page 18, Lines 348 to 351. It would be clearer and more understandable if the authors summarize the hypotheses of their study in the previous sections (e.g., Section 3). Moreover, all research hypotheses should be numbered.

8) Pages 19 and 20, Table 3 and Table 4. The authors should report p-values, rather than only reporting t-values.

Reviewer #2: This paper examines the sociotechnical imagination of autonomous vehicles and compares the differences in cognitive engagement between laboratory and online eye-tracking methods. The research objectives are clear, namely to compare the differences between laboratory and online eye-tracking methods in studying the sociotechnical imagination of autonomous vehicles. The research methods are scientific, the data analysis is thorough, and the results are innovative and practically meaningful. However, there is still room for improvement in aspects such as sample selection and discussion of results.

Simplify the Background Summary: The current background section is relatively lengthy. It can be streamlined to summarize existing research more concisely, highlighting the innovative points of this study. It is recommended to add a concise description of the research questions and hypotheses in the introduction.

The background section provides a detailed overview of research on autonomous vehicle technology and its societal impacts. It is suggested to summarize the main findings of existing research more briefly in the introduction and clearly point out the innovations of this study.

The article describes the experimental design in detail, including specific operational steps for laboratory and online experiments, sample structure, scenario design, etc. Sufficient explanations are also provided for experimental equipment and technical details. However, it needs to more clearly explain the rationality and representativeness of sample selection, especially how to ensure sample diversity and representativeness in online experiments.

By comparing gaze duration, reaction time, and other indicators between laboratory and online experiments, the article analyzes the differences in cognitive engagement between the two methods. It is recommended to add more statistical tests in the data analysis section to verify significant differences in gaze behavior among different groups (such as algorithm-aware and non-aware individuals). Additionally, detailed analysis of different scenario categories can be included.

Clearly describe the specific criteria and steps for sample selection in online experiments to ensure sample diversity and representativeness. Explain how the CINT platform is used to ensure sample representativeness and provide relevant statistical data.

The article discusses the significance of the experimental results, pointing out the differences in data quality and participant behavior between laboratory and online experiments. The conclusion section summarizes the main findings of the study and suggests directions for future research. It is recommended to provide a deeper analysis of the potential reasons for the results in the discussion section, such as the psychological state of participants and the impact of environmental factors under different experimental conditions. Furthermore, the practical application value of the research and its implications for policy formulation can be further discussed.

Some paragraphs in the article are relatively long and can be appropriately segmented to make the content more readable. For certain technical terms, annotations or explanations can be added to help readers better understand.

**Do you want your identity to be public for this peer review?** For information about this choice, including consent withdrawal, please see our Privacy Policy

Reviewer #1: No

Reviewer #2: No

---

## [Author Response · Author response to Decision Letter 1]

14 Oct 2024

Dear editors,

We would like to express our sincere gratitude for the valuable comments on our article ‘Investigating Sociotechnical Imaginaries of Autonomous Vehicles: Comparing Laboratory and Online Eye-Tracking Methodology’.

The reviewers’ comments have helped us to significantly improve our article. Please find attached our revised article and the comments on the reviewers’ suggestions. We have marked the changes in the article using ‘Track changes”, and our responses to reviewers’ comments are below.

Thank you for considering our article for publication! We are looking forward to hearing soon about you!

Yours sincerely,

Mergime Ibrahimi

Anu Masso

Mauro Bellone

Review #1

This paper focuses on an interesting topic. Based on eye-tracking techniques, the authors investigate sociotechnical imaginaries of autonomous vehicles using both in-lab and online eye-tracking experiments. However, there are several issues to be clarified.

1) Page 4, Line 4. The authors need to review more up-to-date articles in Introduction section. Here are three studies for your information. [1] A day in the life with an automated vehicle: Empirical analysis of data from an interactive stated activity-travel survey.

[2] Understanding Human Drivers' Trust in Highly Automated Vehicles via Structural Equation Modeling. [3] Exploring the association between socio‐demographic factors and public acceptance towards fully automated vehicles: Insights from a survey in Australia.

Response: We are highly thankful for this valuable suggestion. We have reviewed the studies mentioned and incorporated them into the Introduction section. Specifically, we have added references [1], [2], and [3] to the introduction on page 4, lines 64-72, where we highlight recent advancements in automated vehicle research. This addition has strengthened the context and relevance of the study within the current literature.

2) Page 8, Lines 145 to 146. The relationship between driver trust on autonomous vehicles and their attention during autonomous driving is also an important factor for traffic safety and public acceptance. See the reference “Latent Hazard Notification for Highly Automated Driving: Expected Safety Benefits and Driver Behavioral Adaptation.”

Response: We appreciate the reviewer’s insightful comment. While trust in autonomous vehicles has been studied extensively, our paper focuses on the perception of individuals towards autonomous vehicles, particularly how imaginaries are formed based on the experience and information available to them. Perception, in this context, is shaped by external narratives and information, which not only influences, but also plays a role in, the creation of trust. Trust, however, tends to develop more gradually through direct experience, whereas perception is shaped by the information individuals receive and how they interpret it.

That being said, we acknowledge the relevance of the suggested study to our broader discussion on safety and public acceptance which shapes people’s imaginaries and perception. As such, we have included the suggested reference on page 8, lines 183-188, where we elaborate on the relationship between trust and driver behavior during automated driving. This addition provides further depth to the discussion on public acceptance.

3) Page 11, Lines 226 to 227. Why not invite automotive engineers/ scientists for the in-lab experiment? As these groups of people are the developer and tester of autonomous vehicles.

Response: We appreciate the reviewer’s suggestion. In fact, our in-lab participants included engineers and scientists from the field of Computer Science (besides Social Scientists), many of whom have hands-on experience with autonomous vehicles and algorithms. Additionally, several participants have been actively involved in studying and evaluating algorithms, providing further depth to their insights. Their technical expertise in relevant areas allowed us to capture valuable perspectives on both the development and testing processes of autonomous vehicles, which enriched the outcomes of our study. Furthermore, besides engineers and scientists from Computer Science, including Social Scientists in our sample enabled us to capture insights that extended beyond technical considerations to encompass potential societal impacts. We clarified this on page 12, lines 294-302.

4) Page 12, Line 233. The authors should give a description on the list of cities and the corresponding languages.

Response: We appreciate the reviewer’s comment. In response, we added in the endnote (page 44) the cities from which participants were drawn. The experiment itself was conducted in English. Given that our participants are rather digitally aware and literate, along with the global spread of the CINT platform – where panel members are frequently engaged in international studies conducted in English – we believe that language did not pose any significant barriers during the experiment.

5) Page 12, Table 1. The authors should report the respondents' experience of using autonomous driving systems.

Reponse: We appreciate the reviewer’s suggestion, and we agree that this is an important aspect to report. In response, we have added the collected data on participants’ trust in and willingness to try fully automated autonomous vehicles to Table 1 (see page 14). This additional information provides further insight into the respondents’ perception toward autonomous driving systems, which is essential for understanding their perspectives in our study.

6) Page 12, Table 1. How do the authors categorize the educational levels?

Repsonse: We appreciate the reviewer’s comment. In response, we have added a legend in Table 1 (page 14) that explains the categorization of educational levels into Basic, Secondary, and Higher categories. However, for clarity and consistency, we followed international studies such as the European Social Survey. Considering the varying education categories used in RealEye platform and CINT for different cities, we merged some of the categories to ensure efficiency. Therefore, we propose the following simplified categories:

•Basic includes early childhood education, primary education, and lower secondary education.

•Secondary includes upper secondary education and post-secondary non-tertiary education.

•Higher includes short-cycle tertiary education, bachelor’s degrees, master’s degrees, and doctoral degrees.

7) Page 18, Lines 348 to 351. It would be clearer and more understandable if the authors summarize the hypotheses of their study in the previous sections (e.g., Section 3). Moreover, all research hypotheses should be numbered.

Response: We appreciate the reviewer’s thoughtful suggestion. Our study is exploratory in nature, however, we recognize the value of this feedback and have now formulated more detailed research questions and assumptions regarding expected results. These have been included in Section 1 on page 5 to provide clearer guidance on the objectives and focus of the study.

8) Pages 19 and 20, Table 3 and Table 4. The authors should report p-values, rather than only reporting t-values.

Response: For Table 3, we have added a note in the legend indicating that each scenario category showed p < 0.001 between AV and non-AV settings, and a detailed explanation on page 22 in the text lines 479-524. Regarding Table 4, we would like to clarify that the sample size was relatively small, which limited our ability to draw definitive conclusions. Therefore, we have added the explanation in the Table 4 legend and further explanation on page 22 in the text lines 525-532.

Review #2

This paper examines the sociotechnical imagination of autonomous vehicles and compares the differences in cognitive engagement between laboratory and online eye-tracking methods. The research objectives are clear, namely to compare the differences between laboratory and online eye-tracking methods in studying the sociotechnical imagination of autonomous vehicles. The research methods are scientific, the data analysis is thorough, and the results are innovative and practically meaningful. However, there is still room for improvement in aspects such as sample selection and discussion of results. Simplify the Background Summary: The current background section is relatively lengthy. It can be streamlined to summarize existing research more concisely, highlighting the innovative points of this study. It is recommended to add a concise description of the research questions and hypotheses in the introduction. The background section provides a detailed overview of research on autonomous vehicle technology and its societal impacts. It is suggested to summarize the main findings of existing research more briefly in the introduction and clearly point out the innovations of this study.

Response: We appreciate the reviewer’s suggestion to streamline the background section. In response, we have revised this section to provide a more concise summary of existing research. We have also emphasized the innovative aspects of our study to make these contributions clearer. Additionally, we have included a brief overview of our research questions and assumptions for expected results in the introduction, guiding the reader through the study’s purpose (Section 1, page 5).

The article describes the experimental design in detail, including specific operational steps for laboratory and online experiments, sample structure, scenario design, etc. Sufficient explanations are also provided for experimental equipment and technical details. However, it needs to more clearly explain the rationality and representativeness of sample selection, especially how to ensure sample diversity and representativeness in online experiments.

Reponse: We appreciate the reviewer’s comment on the importance of explaining sample selection and representativeness more clearly. In our online experiments, we ensured diversity by selecting participants from cities with varying socio-demographic backgrounds, as well as cities that have hosted AV pilots, based on the AV Initiatives Cities © 2017 Bloomberg Philanthropies list. Additionally, we referred to the CMI Rank 2020 to ensure inclusion of cities with both high and low rankings in terms of technological readiness and development. We have added a table in the manuscript Appendix 1 to present the initial online sample plan and the final sample (page 44).

Additionally, we utilized the CINT platform, which allows us to choose the sample based on the proportion of the national census. This ensures that the online sample is representative of the population in terms of key demographic variables (e.g., age, gender, education level, etc.). By leveraging the platform’s capabilities, we were able to enhance the representativeness and diversity of the sample, providing a reliable basis for our analysis. We have added a further explanation on page 13, lines 308-325, and page 14, lines 340-345. Although our goal was to recruit participants based on their respective city census data, it is important to note that online participants tend to be younger and often possess higher educational backgrounds. This sample skewness could result in an over-representation of views from more educated individuals and an under-representation of older people. We have added such limitations in Section 5.1, page 34, line 804-814.

By comparing gaze duration, reaction time, and other indicators between laboratory and online experiments, the article analyzes the differences in cognitive engagement between the two methods. It is recommended to add more statistical tests in the data analysis section to verify significant differences in gaze behavior among different groups (such as algorithm-aware and non-aware individuals). Additionally, detailed analysis of different scenario categories can be included.

Response: We thank the reviewer for this valuable suggestion. In response, we have added additional statistical tests (Table 5, page 24) to the data analysis section to verify significant differences in gaze behavior between the groups and scenario settings, specifically comparing data&algorithm-informed and uninformed individuals. These additional tests are further elaborated on pages 24, lines 569-576, where we report the significance levels and discuss the findings.

Clearly describe the specific criteria and steps for sample selection in online experiments to ensure sample diversity and representativeness. Explain how the CINT platform is used to ensure sample representativeness and provide relevant statistical data.

Response: We thank the reviewer for highlighting the need for more clarity on sample selection criteria and steps. In response, we have added a detailed explanation of specific criteria and steps for sample selection in the online experiment on page 13, lines 308-325, and outlined how we ensured sample diversity and representativeness in the online experiments on page 14, lines 340-345.

The article discusses the significance of the experimental results, pointing out the differences in data quality and participant behavior between laboratory and online experiments. The conclusion section summarizes the main findings of the study and suggests directions for future research. It is recommended to provide a deeper analysis of the potential reasons for the results in the discussion section, such as the psychological state of participants and the impact of environmental factors under different experimental conditions. Furthermore, the practical application value of the research and its implications for policy formulation can be further discussed.

Response: We thank the reviewer for this valuable suggestion. We sought to control the environment by selecting participants who were able to run the experiment via laptops rather than mobile phones. This decision aimed to reduce movement during the experiment, as individuals are generally more stationary when using laptops, which improves the accuracy of eye-tracking data. At the individual level, however, it was more challenging to monitor environmental conditions and psychological states, as the experiments were conducted online without direct supervision from researchers. We acknowledge this as a limitation, as factors such as distractions, lighting, and participant focus could not be fully controlled. This aspect is discussed as a potential drawback of the online method in comparison to in-lab experiments, page 35, lines 815-819.

Additionally, we have expanded our discussion on the practical applications of the research and its implications for policy formulation, providing more detail on how our findings can inform future developments in AV technologies, page 34, lines 790-797.

Some paragraphs in the article are relatively long and can be appropriately segmented to make the content more readable. For certain technical terms, annotations or explanations can be added to help readers better understand.

Response: We have revised the manuscript to improve readability by segmenting longer paragraphs, making the content easier to follow. Additionally, we have added further explanations for technical terms, specifically, we have added legends in the tables to ensure that all readers can fully understand the concepts discussed. These changes are reflected throughout the text, and we believe they enhance the clarity and accessibility of the paper.

---

## [Decision Letter · Decision Letter 1]

17 Dec 2024

Dear Dr. Ibrahimi,

Thank you for submitting your manuscript to PLOS ONE. After careful consideration, we feel that it has merit but does not fully meet PLOS ONE’s publication criteria as it currently stands. Therefore, we invite you to submit a revised version of the manuscript that addresses the points raised during the review process.

Please address all the reviewer's concern and revise the manuscript again.

We look forward to receiving your revised manuscript.

Kind regards,

Quan Yuan, Ph.D.

Academic Editor

PLOS ONE

Journal Requirements:

Reviewers' comments:

Reviewer's Responses to Questions

**Comments to the Author**

Reviewer #1: All comments have been addressed

Reviewer #2: All comments have been addressed

2. Is the manuscript technically sound, and do the data support the conclusions?

Reviewer #1: Yes

Reviewer #2: Yes

3. Has the statistical analysis been performed appropriately and rigorously?

Reviewer #1: Yes

Reviewer #2: Yes

4. Have the authors made all data underlying the findings in their manuscript fully available?

Reviewer #1: Yes

Reviewer #2: No

5. Is the manuscript presented in an intelligible fashion and written in standard English?

Reviewer #1: Yes

Reviewer #2: Yes

Reviewer #1: (No Response)

Reviewer #2: Through the revision of the article, the quality of the paper has been improved to some extent, but there are still some deficiencies in logic and result analysis. The following are some suggestions for improvement:

1. Is the questionnaire used in P12 a questionnaire that has been studied and demonstrated? If not, is the questionnaire tested for its validity, such as reliability and validity?

2. Check whether the conclusions drawn from the analysis of DAI and DAU on page P25 are correct "For Dai participants, classified as such due to their prior participation in similar studies and presumed familiarity with algorithmic systems the average fixation duration (the depth of fixating in one point) and total fixation duration is higher in AV. For DAI, the t-test revealed that there was no statistically significant difference in the fixation durations between AV and non-AV scenarios, t(7080)=1.95, p=0.051. DAU participants showed the same results longer average fixation duration and total fixation duration longer towards AV. For DAU, the t-test indicated that there was a statistically significant difference in the fixation durations between AV and non-AV scenarios, t(10748)=2.11, p=0.034.”

3. A Two-Way ANOVA is used in P25, but it is not stated whether the data meets the conditions of using variance test, such as whether the data meets normal distribution, etc.

4. For the result part, the paper analyzes the differences between the two methods in cognitive participation. However, for the conclusion of the study, for example, Figure 2 uses a visual attention heat map in a single scene to analyze the driver's attention difference under different conditions, but a single case is not representative. Should we consider counting all the experimental data and analyzing it by means of average attention, so as to enhance the persuasiveness of the results?

5. In the part of result analysis, the study found the differences of personal experience, knowledge in specific fields, social demography, culture, geography and online participants on the decision-making of the algorithm, but it did not classify and analyze what personal experience, demography and culture would have on the results, and the analysis of the results was weak, so it was suggested to add.

**Do you want your identity to be public for this peer review?** For information about this choice, including consent withdrawal, please see our Privacy Policy

Reviewer #1: No

Reviewer #2: No

---

## [Author Response · Author response to Decision Letter 2]

31 Jan 2025

1. If the authors have adequately addressed your comments raised in a previous round of review and you feel that this manuscript is now acceptable for publication, you may indicate that here to bypass the “Comments to the Author” section, enter your conflict of interest statement in the “Confidential to Editor” section, and submit your "Accept" recommendation.

Reviewer #1: All comments have been addressed

Reviewer #2: All comments have been addressed

2. Is the manuscript technically sound, and do the data support the conclusions?

Reviewer #1: Yes

Reviewer #2: Yes

3. Has the statistical analysis been performed appropriately and rigorously?

Reviewer #1: Yes

Reviewer #2: Yes

4. Have the authors made all data underlying the findings in their manuscript fully available?

Reviewer #1: Yes

Reviewer #2: No

Response: We are deeply grateful for this feedback and fully understand the requirements of the PLOS Data Policy. We highly value and support the principles and ideals of open science. To meet these requirements, we have made every effort to make the data accessible to other researchers.

However, to ensure the confidentiality and anonymity of the respondents—as recommended by the research ethics committee—we have taken specific measures to balance the openness of science with the protection of research subjects. This is particularly important given the context of our study: a small-scale study with a limited number of participants who could potentially be identified, and a large-scale study addressing the evolving international regulations concerning platform workers’ rights.

To achieve this balance, we have taken the following concrete steps:

1. Meta-data descriptions: We have prepared thorough metadata descriptions to provide detailed explanations of the methodology used in the study. These metadata are openly available here: https://doi.org/10.5281/zenodo.13919434 .

2. Data access request process: Researchers who wish to reuse the data can request access from the authors. In such cases, a confidentiality agreement will be established between the researchers, in line with the data processing principles required by the research ethics committee.

We are committed to finding solutions that uphold the ideals of open science while protecting the rights and privacy of research participants. The steps highlighted above have also been updated in the PLOS ONE submission system.

5. Is the manuscript presented in an intelligible fashion and written in standard English?

Reviewer #1: Yes

Reviewer #2: Yes

Response: We are grateful for this positive feedback. To meet the highest standards, we have revised the article once again, focusing on grammar, style, and the overall smoothness of the text, making a few minor corrections. However, these corrections are purely linguistic, and the content has only been revised in the specific areas where the reviewers requested changes.

6. Review Comments to the Author

Reviewer #1: (No Response)

Reviewer #2: Through the revision of the article, the quality of the paper has been improved to some extent, but there are still some deficiencies in logic and result analysis. The following are some suggestions for improvement:

1. Is the questionnaire used in P12 a questionnaire that has been studied and demonstrated? If not, is the questionnaire tested for its validity, such as reliability and validity?

Response: We thank the reviewer for this question. We used some socio-demographic variables from the survey, such as participants’ country, education level, readiness to use AVs, and prior participation in similar studies, as part of our analysis framework. These variables were collected to contextualize viewing behavior, and are now explained in Section 3, pages 11-12, lines 332-343.

However, all the variables we collected with the survey were not analyzed or demonstrated in this paper, considering the methodological focus of this study. This article primarily focuses on analyzing viewing behavior, comparing two methods, online and in-lab eye-tracking, to contribute to the gaps in previous studies on social imaginaries about automated decision-making systems. Therefore, due to the space limitations of this article, the more in-depth analysis of associations between viewing behavior and background variables will remain the focus of future research. We are very thankful for this suggestion, and we have now explained this aspect in greater detail in the text. You can find the updated explanation in Section 3, page 11-12, lines 343-349.

In regard to validation, the questions were formulated based on validated instruments and frameworks from prior studies. For example, some questions regarding values toward algorithmic solutions were drawn from previously validated studies e.g. Masso, Kaun & van Noordt in 2024. Additionally, we conducted qualitative studies in earlier phases of our research (e.g., Masso & Kasapoglu, 2020) and relied on existing frameworks to formulate formalized questions in this survey. Moreover, during in-lab data collection, in-depth introductory and follow-up interviews were conducted, where participants were also asked to reflect on the experiment and the survey. It gave us an additional opportunity to ensure the reliability of the developed study instruments through cognitive testing of the survey instruments and ensure that the respondents’ understanding of the survey instruments corresponded with the researchers’ ideas. Additionally, test trials have been conducted before in-lab and online study. You can find the updated explanation in Section 3 on page 11, lines 320-333, and page 12, lines 354-358 so that it can be reviewed more easily.

2. Check whether the conclusions drawn from the analysis of DAI and DAU on page P25 are correct "For Dai participants, classified as such due to their prior participation in similar studies and presumed familiarity with algorithmic systems the average fixation duration (the depth of fixating in one point) and total fixation duration is higher in AV. For DAI, the t-test revealed that there was no statistically significant difference in the fixation durations between AV and non-AV scenarios, t(7080)=1.95, p=0.051. DAU participants showed the same results longer average fixation duration and total fixation duration longer towards AV. For DAU, the t-test indicated that there was a statistically significant difference in the fixation durations between AV and non-AV scenarios, t(10748)=2.11, p=0.034.”

Response: We thank the reviewer for pointing out the need to confirm the accuracy of our conclusions based on the reported statistical results. To address this, we have revised the conclusions in the text to ensure alignment with the statistical findings. Specifically:

- For the DAI group, we clarified that while there was a trend towards longer fixation durations in AV scenarios, the result was not statistically significant (t(7080) = 1.95, p = 0.051).

- For the DAU group, we confirmed that the fixation durations were statistically significantly longer in AV scenarios (t(10748) = 2.11, p = 0.034), and this conclusion remains consistent with the data.

The revised text now accurately reflects these findings and avoids overgeneralization for the DAI group. This revision is reflected in Section, 4.2.1, page 25, lines 630-636 of the manuscript.

3. A Two-Way ANOVA is used in P25, but it is not stated whether the data meets the conditions of using variance test, such as whether the data meets normal distribution, etc.

Response: Thank you for the highly valuable comment raised. After your review, we have conducted Shapiro-Wilk for fixation time data. The results indicated deviations from normality (p < 0.05), which is expected given the data’s inherent skewness due to its composition from 26 cities. Furthermore, Levene’s test showed significant differences in variances for some fixation time categories, indicating that the assumption of homogeneity of variances is not fully met. Despite this, Two-Way ANOVA is robust to moderate deviations from normality and variance homogeneity, particularly in large and reasonably balanced datasets. In our study, the sample included 508 participants in the DAI group and 768 in the DAU group.

To validate the findings, we conducted the Wilcoxon Signed-Rank Test to compare fixation times between AV and non-AV settings within each group:

- For the DAI group, the difference in fixation times was statistically significant (p = 0.029).

- For the DAU group, the difference was highly significant (p = 0.0001).

These supplementary analyses confirm the robustness of the Two-Way ANOVA results and support the conclusion that fixation times differ significantly across scenario settings and groups. We have added this additional information and detailed explanations in Section 4.2.1, pages 25-26, lines 637-669, and lines 674-685.

4. For the result part, the paper analyzes the differences between the two methods in cognitive participation. However, for the conclusion of the study, for example, Figure 2 uses a visual attention heat map in a single scene to analyze the driver's attention difference under different conditions, but a single case is not representative. Should we consider counting all the experimental data and analyzing it by means of average attention, so as to enhance the persuasiveness of the results?

Response: We thank the reviewer for this observation. To clarify, Figure 2 is intended as an illustrative example, showing the fixation times on that specific scenario aggregated across all DAI and DAU participants who took part in the experiment. It is not presented as a representative analysis of all experimental data. However, Figure 3 (in-lab heatmap) shows aggregate data from DAI and DAU on all the scenarios, and the chosen scenario is only illustrative. Aggregating data from all the scenarios was possible only with our in-lab eye tracker software. We adjusted the manuscript accordingly to highlight this difference more clearly, in Section 4.2.1, page 27, lines 698-702, and in Section 4.2.2, page 28, lines 727-730 of the manuscript.

For a comprehensive analysis, Fig 1 (boxplots) represents the difference between DAI and DAU based on average fixation duration on the left side, and cumulative fixation duration on the right side. Additionally, we have calculated and reported the average attention across categories in Table 3 (online average fixation time per category) and Table 4 (in-lab average fixation time per category). These tables provide a robust summary of the differences in attention between AV and non-AV scenarios. Thus, Figure 2 serves as a visual illustration to complement these detailed analyses, highlighting the observed differences in attention patterns under specific conditions.

5. In the part of result analysis, the study found the differences of personal experience, knowledge in specific fields, social demography, culture, geography and online participants on the decision-making of the algorithm, but it did not classify and analyze what personal experience, demography and culture would have on the results, and the analysis of the results was weak, so it was suggested to add.

Response: We thank the reviewer for highlighting this important point. Our study has assumed, based on prior research, that imaginaries towards algorithmic decision-making may be shaped by diverse factors such as personal experiences, domain-specific knowledge, socio-demographics (Araujo et al. 2020), and cultural and geographic differences (Grzymek & Puntschuh 2019). While our findings suggest that these differences may play a role in shaping perceptions, we did not specifically classify and analyze the impact of individual factors such as personal experience, demography, or culture, as it was out of our scope in this study. However, this is our focus in the next analysis phases. We have added this additional clarification in the manuscript in Section 3, page 12, lines 347-349, and Section 5, page 36, lines 907-909.

To address the geographic variation further and due to the skewness of our online data (Fig 1), we have already analyzed whether the participant’s country impacted the results. The association between fixation times and participants’ countries was assessed using a chi-square test (page 27, lines 693-696), which revealed a statistically significant association (χ2 = 1284.06, df = 60, p < 0.001). This confirms that geographic location is indeed a factor influencing differences in fixation times. This analysis only scratches the surface of understanding that geography may shape perception. Due to the scope limitations of this article, we will explore this aspect in more depth in future studies, as our main aim was to test the suitability of online and in-lab eye-tracking methods in measuring perceptions towards automated decision-making.

We believe that this limitation highlights the need for future studies to delve deeper into the specific roles that these factors play in shaping imaginaries and perceptions. This was highlighted in Section 5.1, page 38, lines 963-970.

---

## [Decision Letter · Decision Letter 2]

6 Apr 2025

Dear Dr. Ibrahimi,

Thank you for submitting your manuscript to PLOS ONE. After careful consideration, we feel that it has merit but does not fully meet PLOS ONE’s publication criteria as it currently stands. Therefore, we invite you to submit a revised version of the manuscript that addresses the points raised during the review process.

Please address the reviewer's new comments and revise the manuscript again.

We look forward to receiving your revised manuscript.

Kind regards,

Quan Yuan, Ph.D.

Academic Editor

PLOS ONE

Journal Requirements:

Reviewers' comments:

Reviewer's Responses to Questions

**Comments to the Author**

Reviewer #1: All comments have been addressed

Reviewer #2: All comments have been addressed

2. Is the manuscript technically sound, and do the data support the conclusions?

Reviewer #1: Yes

Reviewer #2: Yes

3. Has the statistical analysis been performed appropriately and rigorously?

Reviewer #1: Yes

Reviewer #2: Yes

4. Have the authors made all data underlying the findings in their manuscript fully available?

Reviewer #1: Yes

Reviewer #2: Yes

5. Is the manuscript presented in an intelligible fashion and written in standard English?

Reviewer #1: Yes

Reviewer #2: Yes

Reviewer #1: (No Response)

Reviewer #2: The manuscript makes a valuable contribution by providing novel insights into how individuals perceive autonomous vehicles via complementary eye-tracking methodologies. With revisions addressing data transparency, methodological rigor, detailed statistical justification, and a more comprehensive discussion of limitations and implications, the paper would be suitable for publication.

1.The experimental design, integrating both in-lab and online studies, is well-conceived; however, the relatively small sample size in the laboratory setting (N=16) may limit the generalizability of the findings. This limitation should be more explicitly acknowledged and discussed.

2.The discussion section should more thoroughly integrate the study’s findings with existing literature on sociotechnical imaginaries and automated decision-making. In particular, elaborating on the implications for AV design, public policy, and broader societal impacts would add significant value.

3.Providing further justification for the statistical choices and exploring variance assumptions in greater depth would be beneficial.

**Do you want your identity to be public for this peer review?** For information about this choice, including consent withdrawal, please see our Privacy Policy

Reviewer #1: No

Reviewer #2: No

---

## [Author Response · Author response to Decision Letter 3]

22 May 2025

Dear Dr. Quan Yuan,

We would like to express our sincere gratitude for the valuable feedback on our article ‘Sociotechnical Imaginaries of Autonomous Vehicles: Comparing Laboratory and Online Eye-Tracking Methods’.

Please find attached our revised article and responses to the reviewers’ suggestions. We have marked the changes using ‘Track Changes,’ and our responses below are highlighted in blue.

Based on the reviewers’ recommendations, we have made minor changes to the reference list. The additions to the list of references were made with the aim of supporting the claims regarding the article’s methodological contribution and strengthening the discussions in relation to previous work.

Thank you for considering our article for publication. We look forward to your response.

Yours sincerely,

Mergime Ibrahimi

Anu Masso

Mauro Bellone 

Reviewers’ comments:

Reviewer #2: The manuscript makes a valuable contribution by providing novel insights into how individuals perceive autonomous vehicles via complementary eye-tracking methodologies. With revisions addressing data transparency, methodological rigor, detailed statistical justification, and a more comprehensive discussion of limitations and implications, the paper would be suitable for publication.

1.The experimental design, integrating both in-lab and online studies, is well-conceived; however, the relatively small sample size in the laboratory setting (N=16) may limit the generalizability of the findings. This limitation should be more explicitly acknowledged and discussed.

Response: We thank the reviewer for highlighting this important issue. The relatively small sample size (N=16) for our in-lab experiment indeed presents limitations regarding the generalizability of the findings, especially if this were a standalone study. However, the role of the in-lab experiment in this study was to validate and complement our extensive online eye-tracking study. The large-scale study involved a significantly larger and more diverse participant sample. In the case of a small-scale study, the aim was not to obtain a representative sample, but rather to create a purposive or strategic sample (Suri, 2011) in order to ensure as diverse a range of perspectives as possible, allowing for the design of the present study and the testing of the developed method prior to conducting a larger-scale study. In addition, this comparison of different methods aims to contribute to existing methodological discussions, where increasing attention is being paid to online (semi-experimental) studies. This type of comparison will hopefully also provide a meaningful contribution to the design of future eye-tracking studies, both online and onsite.

To address this explicitly, we have clarified this purpose in the Method section (page 13-14, lines 274-291), Discussion sections (page 38, lines 801-804 and page 39 lines 837-840) and Limitation section (page 41-42, lines 887-897) in the manuscript. Furthermore, we add a sentence in the Method section (page 16, lines 340-342) to acknowledge that territorial representativeness remains a challenge in online crowd-sourced studies.

2.The discussion section should more thoroughly integrate the study’s findings with existing literature on sociotechnical imaginaries and automated decision-making. In particular, elaborating on the implications for AV design, public policy, and broader societal impacts would add significant value.

Response: We sincerely thank the reviewer for emphasizing the importance of contextualizing our findings within existing literature on sociotechnical imaginaries and automated decision-making. We have substantially revised the Discussion section (page 36, lines 753-757, and page 38-39, lines 811-824) to weave our empirical results into two key strands of scholarship: (a) sociotechnical imaginaries of automated decision-making and (b) methodological work using crowd-sourced online eye-tracking. We now open the Discussion by situating AV futures within the “dreamscapes of modernity” framework proposed by Jasanoff (2015), the promises to cure the “driver problem” by removing human error (Braun & Randell, 2020), and the “society-in-the-loop” challenge (Rahwan, 2018) and showing our contribution in demonstrating the potential of online crowd-sourced eye-tracking as a scalable method to capture diverse, global perceptions of AV technologies.

Then we framed our methodological contribution (page 39, lines 830-842). By adding references to other online eye-tracking literature and related crowd-sourced studies, we show how our study repurposes crowd-sourced eye-tracking from usability testing to the measurement of public imaginaries towards emerging technologies. Additionally, we have expanded our discussion (page 40-41, lines 848-878) by referencing specific policy frameworks, particularly highlighting the EU AI Act, to illustrate concrete implications for policymakers, AV designers and developers, and broader society.

3.Providing further justification for the statistical choices and exploring variance assumptions in greater depth would be beneficial.

Response: We appreciate the reviewer’s suggestion to clarify and justify our statistical approach more thoroughly. We have accordingly expanded our explanation of statistical methods (page 26-27), including a deeper discussion of variance assumptions related to our analytical choices. More specifically, we have now added an additional explanation to clarify that the ANOVA (lines 537-539) was chosen because it tests both main effects and their interaction and remains reliable with the large data sizes. We reported the effect-size (page 26, line 540 and page 27, line 543) – partial η² values are now given for the group effect (0.009) and the scenario effect (0.0005) to quantify the magnitude of each result. We added assumption checks (page 27, lines 552-556): 1) Shapiro–Wilk on a random 5 000-residual subset (W = 0.663, p <0 .001) is reported, and we briefly justify the 5 000-case subsample as the practical upper limit for the test; and 2) Levene’s test is now presented with its exact statistic (F(3, 17 860) = 24.72, p < .001). To demonstrate that our findings do not hinge on the violated assumptions, we added (page 27-28, lines 556-566) a Welch heteroscedastic two-way ANOVA (HC3 estimator) and a Gamma-log generalized linear model, both of which reproduced the group and scenario main effects and the null interaction. Furthermore, we now reported both versions of the Wilcoxon test (page 28, lines 568-573) to give a transparent view of the data’s robustness 1) fixation-event level (as in the original draft) maximizes statistical power by using every gaze event as a pair; this yields p = 0.029 for DAI and p = 0.0001 for DAU, and 2) participant-level means aggregate those events within individuals, eliminating the risk of pseudo-replication and providing a more conservative inference; here the DAI contrast becomes a non-significant trend (p = 0.105) whereas the DAU contrast remains significant (p = 0.044). Including both analyses demonstrates that the overall scenario effect is stable, while also signaling to readers how the choice of statistical unit affects the strength of the evidence, an important consideration for future replication and meta-analysis.

These insertions provide thorough statistical justification and variance exploration, while keeping the substantive conclusions unchanged.

---

## [Decision Letter · Decision Letter 3]

15 Oct 2025

Sociotechnical Imaginaries of Autonomous Vehicles: Comparing Laboratory and Online Eye-Tracking Methods

PONE-D-24-17301R3

Dear Dr. Ibrahimi,

We’re pleased to inform you that your manuscript has been judged scientifically suitable for publication and will be formally accepted for publication once it meets all outstanding technical requirements.

Kind regards,

Quan Yuan, Ph.D.

Academic Editor

PLOS ONE

Additional Editor Comments (optional):

Reviewers' comments:

Reviewer's Responses to Questions

**Comments to the Author**

Reviewer #1: All comments have been addressed

2. Is the manuscript technically sound, and do the data support the conclusions?

Reviewer #1: Yes

3. Has the statistical analysis been performed appropriately and rigorously?

Reviewer #1: Yes

4. Have the authors made all data underlying the findings in their manuscript fully available?

Reviewer #1: Yes

5. Is the manuscript presented in an intelligible fashion and written in standard English?

Reviewer #1: Yes

Reviewer #1: The authors have adequately addressed all of my comments. Therefore, the manuscript can be accepted now.

**Do you want your identity to be public for this peer review?** For information about this choice, including consent withdrawal, please see our Privacy Policy

Reviewer #1: No

---

## [Editor Report · Acceptance letter]

PONE-D-24-17301R3

PLOS ONE

Dear Dr. Ibrahimi,

I'm pleased to inform you that your manuscript has been deemed suitable for publication in PLOS ONE. Congratulations! Your manuscript is now being handed over to our production team.

Kind regards,

on behalf of

Dr. Quan Yuan

Academic Editor

PLOS ONE